# Physical Activity-Dependent Regulation of Parathyroid Hormone and Calcium-Phosphorous Metabolism

**DOI:** 10.3390/ijms21155388

**Published:** 2020-07-29

**Authors:** Giovanni Lombardi, Ewa Ziemann, Giuseppe Banfi, Sabrina Corbetta

**Affiliations:** 1Laboratory of Experimental Biochemistry & Molecular Biology, IRCCS Istituto Ortopedico Galeazzi, 20161 Milan, Italy; banfi.giuseppe@fondazionesanraffaele.it; 2Department of Athletics, Strength and Conditioning, Poznań University of Physical Education, 61–871 Poznań, Poland; ziemann.ewa@gmail.com; 3Faculty of Medicine and Surgery, Vita-Salute San Raffaele University, 20132 Milan, Italy; 4Endocrinology and Diabetology Service, IRCCS Istituto Ortopedico Galeazzi, 20161 Milan, Italy; sabrina.corbetta@unimi.it; 5Department of Biomedical, Surgical and Dental Sciences, University of Milan, 20122 Milan, Italy

**Keywords:** physical activity, skeletal muscles, PTH, calcium, vitamin D, phosphate, irisin, osteocalcin, hyperparathyroidism, hypoparathyroidism

## Abstract

Exercise perturbs homeostasis, alters the levels of circulating mediators and hormones, and increases the demand by skeletal muscles and other vital organs for energy substrates. Exercise also affects bone and mineral metabolism, particularly calcium and phosphate, both of which are essential for muscle contraction, neuromuscular signaling, biosynthesis of adenosine triphosphate (ATP), and other energy substrates. Parathyroid hormone (PTH) is involved in the regulation of calcium and phosphate homeostasis. Understanding the effects of exercise on PTH secretion is fundamental for appreciating how the body adapts to exercise. Altered PTH metabolism underlies hyperparathyroidism and hypoparathyroidism, the complications of which affect the organs involved in calcium and phosphorous metabolism (bone and kidney) and other body systems as well. Exercise affects PTH expression and secretion by altering the circulating levels of calcium and phosphate. In turn, PTH responds directly to exercise and exercise-induced myokines. Here, we review the main concepts of the regulation of PTH expression and secretion under physiological conditions, in acute and chronic exercise, and in relation to PTH-related disorders.

## 1. Introduction

Exercise perturbs homeostasis. Acting as a stressor, the more intense and the longer the exercise bout, the stronger the perturbation. Acute physical activity induces a hormone response that re-establishes homeostasis. The endocrine response to acute exercise occurs over multiple phases and its magnitude depends on the work volume and intensity. In contrast, chronic exercise (training) leads to an adaptation to this stimulus and attenuates the body’s response to acute exercise intensity and volume, without abolishing it, however. At an intensity and volume above an individual’s ability to compensate, maladaptation can lead to an abnormal hormone response and functional failure, which can manifest in overreaching and overtraining syndromes [1].

Calcium-phosphate homeostasis is fundamental for the functioning of all cell types in the body, which is why the levels of calcium and phosphorus in the blood are tightly controlled within a very narrow range. Calcium and phosphorous are also essential for the functioning of striated skeletal muscle and cardiac muscle cells, as well as for neuronal and neuromuscular activity. The homeostatic control of the two elements is important for physical activity and exercise performance. The homeostasis of calcium and phosphorous is maintained by the regulation of the entrance gates (the intestines), the exit gates (the kidneys), and the storehouse (the skeleton). The regulation signal is generated by parathyroid hormone (PTH) and other hormones [2].

PTH is primarily expressed by four small glands located behind the thyroid gland, the parathyroid glands, which are devoted to the control of calcium homeostasis [3]. Despite its importance in physiology and its known role in diverse primary and secondary diseases (i.e., hyperparathyroidism and hypoparathyroidism), PTH has been far less studied in exercise endocrinology than calcium-phosphate metabolism. In the present review, we report on recent research into PTH and PTH-dependent calcium-phosphate response to acute and chronic exercise.

## 2. Physical Activity, Exercise, and Training: Terminology and Explanation

In popular parlance, the terms and concepts proper to exercise physiology are commonly misused. A clear explanation of the biological mechanisms involved in exercise-induced homeostatic perturbation and the body’s response is therefore warranted. The terms physical activity, exercise, and training are often interchangeably and sometimes incorrectly used. Physical activity broadly refers to any bodily movement that is produced by skeletal muscles and that results in energy consumption. It is intrinsically associated with activities of daily living; except for involuntary actions, the amount of physical activity an individual performs is subjective. Exercise refers to physical activity that is planned, structured, and repetitive. The two terms are often used synonymously. However, exercise is distinguished by the aim to maintain or improve physical fitness [4]. A single bout of exercise is defined as acute exercise, independent of an individual’s physically active status (e.g., an isolated exercise bout in a sedentary subject vs. an exercise bout within a long-term exercise program in a physically active subject), while exercise bouts continued over time constitute training (exercise training).

Muscle action can be distinguished as static (isometric) and dynamic (isotonic). During isometric action, the muscle generates force but its length remains unchanged: external resistance, the weight of a held object for example, exceeds the force generated by the muscle. Additionally, since no movement occurs, no work is performed despite the expenditure of energy. Work is performed during isotonic action: in concentric contraction, the muscle generates a force that overcomes external resistance and causes the muscle fibers to shorten; in eccentric contraction, the sarcomere shortens and the muscle lengthens due to the opposite movement of the external resistance [5].

Categorized by type of metabolism, exercise and training can be defined as: (1) resistance (or strength needed for weight lifting, discus, or hammer or javelin throw) or short-term power or explosive activity dependent mainly on anaerobic metabolism; and (2) endurance, mid-to-long-term activity dependent mainly on aerobic metabolism (e.g., distance running, road cycling, swimming, and triathlon) [6].

Table 1 presents the characteristics of physical activity and exercise.

## 3. Physiology of PTH and Calcium-Phosphate Metabolism

### 3.1. Synthesis, Secretion, Metabolism, and Mechanisms of Action of the Parathyroid Hormone

PTH is expressed and secreted by the chief cells and the oxyphil cells of the parathyroid glands and regulated by a gene on the short arm of chromosome 11 [7]. It is expressed as 115-amino-acid pre-pro-PTH, which is then cleaved in the Golgi apparatus to yield the mature 84-amino-acid PTH and then stored in granules [8]. *PTH* mRNA transcription and stability are negatively regulated by extracellular calcium concentrations ([Ca^2+^]_e_) by binding to the calcium-sensing receptor (CASR), which displays half-maximal inhibition at about 1 mmol/L [Ca^2+^]_e_ [9,10]. Moreover, the rate of change in [Ca^2+^]_e_ drives the robustness of the response (i.e., the more rapid the drop in [Ca^2+^]_e_, the greater the magnitude of PTH secretion) [8]. In normocalcemic conditions, only 20% of PTH circulates in its intact form (full-length sequence comprises of aa 1–84, thereafter indicated as PTH(1–84)), while 80% is present as inactive fragments [11]. Increased [Ca^2+^]_e_ stimulates degradation of PTH within the parathyroid cell, which generates biologically inactive C-terminal fragments (e.g., comprising of the aa sequence from 34 to 84 (PTH(34–84)) or from 37 to 84 (PTH(37–84))) [12]. PTH is also degraded by the liver and cleared by the kidney [13]. Though full-length PTH is the prevalent bioactive circulating form, all PTH-derived peptides retaining the N-terminal have long been recognized as biologically active [14]. Low [Ca^2+^]_e_ stimulates parathyroid cell proliferation and total secretory capacity, while hypercalcemia may induce parathyroid gland hypotrophy, though it is less associated with hyperplasia [15]. PTH secretion exhibits seasonal and circadian fluctuations synchronous with changes in serum calcium, phosphate, and bone turnover. In addition, an ultradian rhythm exists that comprises of seven pulses per hour, accounts for 30% of basal PTH release, and is highly sensitive to changes in ionized calcium. Acute hypocalcemia induces a selective, several-fold increase in pulse frequency and amplitude, whereas hypercalcemia suppresses the pulsatile secretion component [16].

*PTH* gene transcription is inhibited by 1α,25-dihydroxyvitamin D (1α,25-(OH)_2_D), the active form of vitamin D, which also inhibits parathyroid cell proliferation [15,17]. Phosphate, which is closely interrelated with Ca^2+^ metabolism, plays a role in modulating *PTH* gene transcription and protein secretion. The endocrine regulation of phosphate is carried out by PTH itself, 1α,25-(OH)_2_D, and recently discovered bone-derived phosphatonin, fibroblast growth factor (FGF)23 [18]. Mainly secreted by osteocytes, FGF23 plays a central role. Increases in circulating phosphate over days, but not acute increases over hours, upregulates FGF23 secretion, resulting in a phosphaturic effect. Simultaneously, renal 1α,25-(OH)_2_D production falls, resulting in decreased gut absorption of phosphate and calcium. The overall effect is to prevent hyperphosphatemia and ectopic mineralization. Diurnal variation in serum phosphate is not associated with increases in serum FGF23. This contrasts with diurnal variation in serum Ca^2+^, which is inversely correlated with changes in serum PTH levels. FGF23 regulates phosphate metabolism, whereas PTH and 1α,25-(OH)_2_D regulate both phosphate and Ca^2+^ metabolism. Circulating phosphate concentrations do not directly affect PTH secretion, although high dietary phosphorus intake and oral phosphate supplements do so indirectly by reducing the [Ca^2+^]o concentration. The three hormones (PTH, 1α,25-(OH)_2_D, and FGF23) modulate one another’s secretion: FGF23 decreases PTH secretion, 1α,25-(OH)_2_D decreases PTH secretion and increases FGF23 secretion, and PTH increases FGF23 secretion [19].

Finally, PTH synthesis and secretion are modulated by the transforming growth factor (TGF)-α, prostaglandins (PGs), and cations such as lithium. Figure 1 illustrates the regulation of PTH expression and secretion.

PTH binds through its N-terminal domain a specific class II G-protein coupled receptor (GPCR) expressed by the target tissues, the PTH/PTHrP (PTH related peptide) receptor type 1 (PTHR1) [20,21]. PTHR1 is also targeted by PTHrP, a peptide sharing homology with the N-terminal portion of PTH. Unlike PTH, PTHrP explicates its action in an autocrine/paracrine fashion. Based on PTHR1 expression patterns, the main PTH targets are the bone and the kidney [22]. PTHR1 is coupled with different G-protein classes, resulting in the activation of several intracellular pathways. Gαs activates adenylate cyclase (AC), which results in the synthesis of cyclic AMP (cAMP) and the activation of protein kinase (PK) A. Activated PKA phosphorylates transcription factors, including the cAMP-response element-binding (CREB) protein. CREB phosphorylation regulates its interaction with the cAMP-responsive element (CRE) on DNA and the transcription of target genes. Gαq activates phospholipase (PL) C, which cuts a membrane-enriched phospholipid, phosphatidylinositol-(4,5)-bisphosphate (PIP_2_), into diacylglycerol (DAG) and inositol-(1,4,5)-trisphosphate (IP_3_). By binding its receptor in the endoplasmic reticulum, IP_3_ activates the receptor-gated Ca^2+^ channel and the release of calcium into the cytoplasm from the endoplasmic reticulum stores. The derived Ca^2+^ spike allows the translocation of PKC to the plasma membrane, where DAG, released by PLC, activates PKC. Gα12/Gα13 activates PLD [18]. Figure 2 illustrates the PTH-induced signaling pathways.

PTH binding to its cognate receptor induces desensitization, which is initiated by phosphorylation of a serine residue at the C-terminus of PTHR1 mediated by GPCR kinase (GRK). This reaction culminates in the sequestration of the receptor within clathrin-coated endocytic vesicles, where the receptor is still able to activate Gs-mediated signaling and its degradation or recycling to the membrane [23,24].

Finally, the intracellular tail of PTHR1 interacts with adaptor proteins, such as those belonging to the Na/H exchanger regulatory protein factor group (NHERFs) that protect the receptor from downregulation, stimulates its interaction with Gq (stimulation of the PLC-PKC pathway) and inhibits its interaction with Gs (inhibition of the AC-PKA-CREB pathway). Modulation by the NHERFs and activation of the downstream pathway depend upon their expression profile in tissues: in osteoblasts and kidney tubule cells, both Gs- and Gq-dependent pathways can be activated [25,26,27,28,29].

### 3.2. Parathyroid Hormone and Calcium-Phosphate Homeostasis

The action of PTH is devoted mainly to maintaining [Ca^2+^]_e_ within the normal range. A [Ca^2+^]_e_ decrease induces the secretion of PTH from the parathyroid glands. In response, reabsorption of Ca^2+^ (and Mg^2+^) in the kidney tubule is increased, while P_i_ (and HCO_3_^-^) reabsorption is inhibited. PTH and hypocalcemia stimulate the AC-PKA-cytochrome P27B1 (CYP27B1) pathway-mediated hydroxylation of 25-hydroxyvitamin D (25-(OH)D) into the active form (1α,25-(OH)_2_D). In the small intestines, 1α,25-(OH)_2_D increases Ca^2+^ (and also P_i_) absorption. In the bone, PTH rapidly enhances the turnover rate, thus mobilizing Ca^2+^ and P_i_. Once [Ca^2+^]_e_ is restored, PTH expression and secretion and activation of vitamin D are again inhibited [30].

#### 3.2.1. The Kidney

Calcium reabsorption in the kidney takes place along the whole nephron, but the major part is taken up in the proximal tubule where most of the solutes and water are resorbed [31]. However, the effects of PTH on calcium reabsorption are limited to the distal portion of the nephron. In the cortical, thick ascending Henle’s loop, 20% of filtered Ca^2+^ is reabsorbed through activation of PTHR1, which increases the activity of the Na/k/Cl cotransporter, allowing reabsorption of NaCl and, in turn, the paracellular reabsorption of Ca^2+^ (and Mg^2+^). In contrast, when [Ca^2+^]_e_ rises, paracellular calcium resorption is inhibited by the activation of CASR [32,33,34]: 15% of the filtered calcium is reabsorbed in the distal convoluted tubule via a transcellular route mediated by the apically expressed, highly selective Ca^2+^ channel transient receptor potential cation channel subfamily V member 5 (TRPV5). Once in the tubule cell, the calcium is translocated to the basolateral membrane by carriers (e.g., calbindin-D28k) and is eliminated by the sodium/calcium exchanger (NCX)1. The PTH-dependent activation of PKC controls the expression and the activity of both TRPV5 and NCX1, as CaSR does for TRPV5 [35,36,37].

Filtered Pi is reabsorbed only in the proximal tubule, where the epithelial cells express two sodium-dependent phosphate transporters (NaPi-IIa and NaPi-IIc) on the apical side [38]. In this section of the nephron, PTHR1 is expressed [39,40] and associated with both the apical membrane, where it is coupled to the NHERF-dependent PLC-PKC signaling pathway [41], and the basolateral membrane, where it is coupled to the AC-PKA pathway [42]. In both pathways, stimulation by PTH results in downregulation of NaPi-IIa/IIc and reabsorption of Pi [43,44]. Most of the cAMP generated by the activation of tubular PTHR1 is eliminated with the urine and it can be considered a reliable marker of the PTH function [45].

#### 3.2.2. Bone

PTHR1 is expressed by the bone cells of the osteoblastic lineage (including osteoprogenitors, lining cells, immature and mature osteoblasts, and osteocytes) as well as by mesenchymal stem cells (MSC). PTH stimulation enhances the cell proliferation rate, commitment towards osteoblastic differentiation, osteoblastic activity, bone matrix deposition, and mineralization. PTH can also stimulate the expression of the tumor-necrosis factor (TNF)α-related ligand of the receptor activator of nuclear factor κB (RANKL), a stimulating factor for osteoclast differentiation and activity that expresses the receptor for RANKL (RANK) and inhibits osteoprotegerin (OPG), a decoy receptor for RANKL. Osteoclast activation causes bone resorption and the release of Ca^2+^ and P_i_ from the resorbing bone. PTH stimulates bone turnover, but the direction of this stimulus toward formation (anabolic) or resorption (catabolic) depends on other factors.

The anabolic role of PTH in bone has been demonstrated in mice in which PTH knockout (KO) was noted to reduce trabecular bone mass and the number of metaphyseal osteoblasts in fetal and neonatal life [46] and in adult life when PTH is needed for fracture healing [47]. In humans, the role of PTH in the bone becomes evident when PTH physiology is more or less severely affected: hypoparathyroidism (parathyroid gland hypofunction associated with reduced secretion of PTH), hyperparathyroidism (parathyroid gland hyperfunction associated with inappropriate secretion of PTH), and PTH administration are all associated with dysregulation of bone remodeling.

PTH stimulates cell proliferation through the cAMP-mediated induction of c-fos in stromal cells and osteoblasts and MAPK cascades in osteoblasts [48,49,50]. cAMP signaling may also induce Runx2 (also known as Cbfa1), an early transcriptional regulator of osteoblast differentiation [51]. Osteoblast differentiation is also driven by the PTH-induced activation of Tmem119 (Smad3-related factor), which stimulates the bone morphogenic protein (BMP)-Runx2 pathway [52]. PTH-induced osteoblastic differentiation implies the exit from the cell cycle dependent on the inhibition of cyclin D1 expression and the induction of cell cycle inhibitors (p27Kip1 and p21Cip1) [53,54], as well as the inhibition of proapoptotic factors (Bad), the induction of antiapoptotic factors (Bcl) [55], and the increased efficiency of DNA repair [56].

PTH also affects the expression of factors interfering with osteoblast differentiation that are produced by other cells. It inhibits the expression of the unloading-stimulated osteocyte-derived osteoblastogenic inhibitor sclerostin (Sost) [57,58]. PTH induces and stimulates the release of insulin-like growth factor (IGF)-1, an inducer of the osteoblastic pool and osteoblast differentiation [59], and the release of FGF2, which has proproliferative and antiapoptotic effects on osteoblasts [60].

Besides the anabolic effects on bone, when [Ca^2+^]_e_ is decreased, PTH released from the parathyroid glands induces bone resorption or, better, causes an imbalance in bone turnover towards resorption. This effect seems to be limited to cortical bone, though when blood PTH levels are elevated, the trabecular compartment may also be involved. PTH induces the expression of RANKL in osteoblastic cells and downregulates OPG expression, which stimulates osteoclastogenesis and osteoclast resorption [61,62]. Along with calcium and phosphate, bone matrix resorption causes the release of other factors buried in the matrix during its synthesis (e.g., TGF, BMPs, FGFs and IGF), which, in turn, induces osteoblastogenesis and osteoblastic activity [18]. Furthermore, activated osteoclasts directly recruit osteoblast and osteoblast precursors at the resorption pit by secreting factors (e.g., ephrin, sphingosine-1-phosphate (S1P) and BMP6) [63,64,65,66]. The PTH-mediated stimulation of bone resorption is one way by which the body increases the osteoblast pool and stimulates osteoblastic activity and bone anabolism [18].

In patients with osteoporosis, PTH (teriparatide and abaloparatide) is administered intermittently; there is an early phase of enhanced bone formation without resorption (anabolic window), followed by a phase of enhanced turnover (formation and resorption) [67,68]. The temporary extension of the anabolic window allows for a net bone accrual and a gain of up to + 20% in trabecular and endocortical bone, while periosteal bone is mainly subject to the modeling effects of prolonged treatment (less than 10% of bone gain) [69,70,71,72,73].

## 4. Evidence for a Role of PTH in Conditions of Abnormal PTH Secretion and Movement-Associated Function

Mineral metabolism is regulated by the interaction of factors with specific actions that may be indistinguishable from one another. For the present review, we focused on the effects induced by PTH. Empirical evidence for the role of PTH and the regulation of calcium metabolism is taken from conditions in which this system is altered. Parathyroid diseases comprise of conditions in which there is an imbalance in calcium and phosphate homeostasis due to alterations in PTH secretion or action. Besides its effects on calcium and phosphate homeostasis, abnormal PTH secretion impacts on energy metabolism. Epidemiological studies have demonstrated an association between circulating PTH levels, a negative metabolic profile, and increased cardiovascular risk [74]. However, PTH secretion is also influenced by a number of physiological, pathological, and treatment-related factors.

The range of circulating PTH levels is narrower in healthy children and adolescents than in adults, often < 40 pg/mL compared to the upper limit of 65 pg/mL in adults. Circulating PTH levels also tend to increase with age. In both women and men, ageing *per se*, independent of changes in vitamin D economy or renal function, is associated with an increase in the integrated PTH secretory response to changes in serum calcium, though no alterations in the Ca/PTH set-point are present [75]. Circulating PTH levels can be affected by systemic metabolic disorders such as arterial hypertension and insulin resistance/diabetes. Elevated circulating PTH levels, owing to the increase in intracellular calcium, can impair mitochondrial function and ATP production and contribute to oxidative stress, as well as inflammatory states and, ultimately, to cardiomyocyte necrosis. The interplay between PTH, FGF23, and aldosterone is also detrimental for the cardiovascular system and is involved in endothelial dysfunction [76]. Chronic elevation of PTH levels, as in hyperparathyroidism, has been associated with insulin-resistance and diabetes. Finally, some drugs can affect serum PTH levels: osteoporotic treatment with bisphosphonate and denosumab, two anti-resorptive agents, raises PTH levels, while treatment with teriparatide decreases circulating PTH levels. Canaglifozin and dapaglifozin, SGLT2 inhibitors for the treatment of diabetes, have been reported to increase PTH levels in association with increased serum phosphate and FGF23 [77,78]. In a large community-based cohort with normal kidney function, use of thiazide diuretics was associated with lower PTH, whereas loop diuretics and dihydropyridine calcium-channel blockers were associated with higher PTH [79]. In patients with primary hyperparathyroidism and in hypertensive patients, treatment with angiotensin-converting enzyme inhibitors decreased serum PTH levels [80,81].

### 4.1. Hypoparathyroidism

Hypoparathyroidism results from the destruction or dysfunction of the parathyroid glands and leads to a permanent or transient lack of PTH, most commonly caused by thyroidectomy-associated inadvertent damage/removal of the parathyroid glands [82], whereas it is rarely associated with parathyroidectomy (PTX), autoimmunity, Di-George syndrome, activating mutations of the *CASR* gene, and inactivating mutations of the glial cell missing 2 (*GCM2*) transcription factor or *PTH* genes. Diagnosis is based on low serum albumin-corrected calcium concentration and inappropriately low/undetectable PTH [83]. Hypocalcemia is associated with low circulating levels of 1α,25-(OH)_2_D and FGF23. Hypocalcemia and related complications are treated with oral calcium, active vitamin D supplements, and subcutaneous injection of recombinant PTH 1-84 [84]. The clinical presentation of hypoparathyroidism can be also sustained by PTH resistance, mostly due to inactivating mutations of the *GNAS1* gene, a disorder known as pseudohypoparathyroidism. Pseudohypoparathyroidism differs biochemically from hypoparathyroidism by the occurrence of detectable and often elevated serum PTH levels.

Insufficient PTH secretion or function results in signs and symptoms arising from hypocalcemia, which interferes with muscle contraction and nerve conduction: paresthesia, tingling sensation in the mouth, hands and feet, cramps, and tetanic muscle spasms of the hands and feet. Hypocalcemia can also cause seizures, severe arrhythmias, laryngospasm, dyspnea, and bronchospasm [85]. Hypocalcemia-dependent neuromuscular irritability manifests as tetany at the fingertips, tingling sensation at the mouth and toes, and carpo-pedal spasms. Mechanical/ischemic stimulation above the hypocalcemia-dependent decreased impulse threshold may evoke muscle spasms of Chvostek (facial) and Trousseau (ulnar) signs, respectively. Muscle weakness and/or stiffness and increased circulating muscle enzyme activity (e.g., creatine kinase) are common [86]. Neuromotor impairment (prevalence 4–12%) results in parkinsonism (bradykinesia, mask-like face, tremor, mixing posture) and extrapyramidal (choreiform movement, hemiballismus) or cerebellar signs (ataxia, dysarthria) [87]; also common are cognitive (concentration, memory) and mood (e.g., anxiety) disturbances [88,89]. The underlying mechanisms have been identified in cerebral and cerebellar calcifications, mainly in the basal ganglia [87]. Symptoms improve with correction of hypocalcemia (calcium and vitamin D supplementation) [86,87,88,89]; however, PTH besides hypocalcemia may play a direct role since PTHR is expressed in various areas of the brain [88,89].

Hypocalcemia causes electrocardiographic abnormalities (prolonged QTc interval), despite possibly preserved cardiac function, and can be associated with dilated cardiomyopathy, global hypokinesia, reduced ejection fraction, and, rarely, with treatment-resistant heart failure. These signs can be effectively reversed by correcting hypocalcemia [90,91]. Low serum [Ca^2+^] associated with low [Ca^2+^]_e_ regulates myocardial excitation-contraction coupling: Ca^2+^ enters through L-type calcium channels, stimulates the release of Ca^2+^ from the sarcoplasmic reticulum (Ca^2+^-induced Ca^2+^ release (CICR)), which binds troponin C and activates the contraction [92,93]. A reduced extracellular-to-intracellular Ca^2+^ gradient attenuates CIRC, impairs contraction [91], and leads to fatigue [94,95].

In the bone, hypoparathyroidism is associated with higher bone mineral density (BMD) compared to age- and sex-matched healthy subjects, with about a 1-unit gain in dual energy X-ray absorptiometry (DXA)-determined Z- and T-scores, especially at the lumbar spine [96], and greater radius and tibia cortical BMD, as determined by high-resolution peripheral quantitative computed tomography (pQCT). Slow bone metabolism is marked by low-to-normal levels of bone turnover markers [97] and a 54–80% decrease in resorption and formation at trans-iliac bone biopsy [98]. MicroCT reveals trabecular involvement with greater number, thickness, and number of connections between the trabeculae [99]. Hypoparathyroidism is associated with increased fracture risk, especially of the upper extremities [91].

Among the other manifestations of hypoparathyroidism (gastrointestinal, ophthalmic, cutaneous, and dental) [85], gastrointestinal symptoms due to muscular dysfunction include abnormal peristalsis, which results in constipation and abdominal cramps.

### 4.2. Hyperparathyroidism

Primary hyperparathyroidism (PHPT), the third most common endocrine disease, after type 2 diabetes (T2D) and thyroid disorders (prevalence 1–3:1000 in the general population, up to 20:1000 in postmenopausal women) [100,101]; the hallmark features are hypercalcemia and increased or inappropriately normal levels of PTH [102]. PHPT can be asymptomatic, with increased or normal calcium levels [103]. Generally, PHPT is caused by a single, benign parathyroid adenoma; a fifth of cases are due to multiple glandular hyperplasia (including the familial forms) and rarely to malignant parathyroid carcinoma [104,105].

Vitamin D deficiency frequently occurs in patients with PHPT and affects the biochemical characteristics of PHPT. Recent guidelines for the clinical management of PHPT recommend correction of vitamin D deficiency when diagnosing PHPT [106]. PHPT patients present with elevated circulating FGF23 levels, whose role in PHPT is not completely elucidated [107].

Skeletal findings (subperiosteal bone resorption, distal clavicle osteolysis, salt and pepper skull, bone cysts, brown tumors, and very low BMD [108]) were considered pathognomonic before the event of routine determination of serum calcium. Due to earlier diagnosis and milder vitamin D deficiency, the radiological manifestations are no longer seen (asymptomatic PHPT) in Western countries [109,110] unlike in Eastern countries (e.g., China and India) where they are still common [111,112]. In asymptomatic PHPT, BMD is typically decreased at the forearm, intermediate at the hip, and unchanged at the lumbar spine [108], with a pattern opposite that of postmenopausal osteoporosis [101]. Bone turnover is histomorphometrically and biochemically increased mainly in the cortical compartment [108] and less in the trabecular compartment due to the more prominent action of excessive PTH at the corticotrabecular junction. PHPT is associated with increased risk of vertebral and non-vertebral fracture, as predicted by the DXA-obtained trabecular bone score (TBS) [113,114,115,116,117], which, despite partial preservation of the trabecular compartment, is noted to be low in PHPT patients [118,119,120]. Bone [Ca^2+^] is low in PHPT due to the accelerated turnover, as are the collagen crosslinks. PTX achieves a long-term improvement in BMD at all sites, and in cortical and trabecular microstructure [108].

PHPT causes hypercalciuria, silent or overt microlithiasis and nephrolithiasis, and related complications (polyuria, urinary tract infections, hydronephrosis, impaired kidney function, and kidney failure), also in asymptomatic subjects [102,121,122], with a similar prevalence between normocalcemic and hypercalcemic patients [123]. Calcium phosphate and calcium oxalate are the most common stone forms, while calcium urate formations are uncommon; precipitation of crystals may be promoted by increased urinary phosphate, oxalate, and sodium, decreased urine citrate concentrations and proteinuria [102]. In silent stone formers, nephrolithiasis is associated with low 25-(OH)D concentrations [124]. Vitamin D supplementation reduces serum PTH, without causing hypercalcemia and hypercalciuria, thus reducing the risk of nephrolithiasis [125,126]. Nephrolithiasis, which is more common in males [127], is an indication for PTX [106,128,129]. Chronic kidney disease (CKD), a long-term complication of PHPT with a prevalence of 16–17% [130], is the expression of a severe phenotype and carries an increased risk of morbidity and mortality [102] due to its association with hypertension, decreased BMD, and worse diabetic status [131]. PTH levels are associated with the degree of renal impairment in PHPT [132], and PTX prevents further kidney impairment [133]. CKD and related complications are seen also in other forms of hyperparathyroidism (e.g., multiple endocrine neoplasia type 1 (MEN1), type 2A (MEN2A), and type 4 (MEN4), parathyroid carcinoma, and gestational hyperparathyroidism) [102,134,135,136].

PHPT patients are at higher risk of cardiovascular disease compared to the general population (odds ratio (OR) 1.7–2.5) [137]. When asymptomatic, cardiovascular abnormalities are not associated with increased morbidity and mortality and elevated PTH, except for hypercalcemia, which seems to be associated with an increased risk of fatal and non-fatal cardiovascular disease in mild PHPT [138,139].

Among other nonspecific symptoms, neuromuscular and articular manifestations have been reported in PHPT patients [140], but the epidemiology remains uncertain. Neuromuscular involvement is reported in 13–93% of PHPT patients [140,141]. PTH is also involved in metabolic alterations that may lead to joint impairment [142]. It is independently associated with serum uric acid [143]; PHPT may be associated with an increased risk of gout [140,144]. Calcium phyrophosphate deposition disease, known as articular chondrocalcinosis, is the most common articular finding in PHPT, with a possible direct implication for PTH. Since joint pain is almost invariably reported by PHPT patients [140,144,145], it should be remembered that osteoarthritis is very common in the age range of patients diagnosed with PHPT [142]. Less common manifestations include Achilles tendon rupture, Jaccoud-like arthropathy, sacral insufficiency fractures, and arthritis associated with fever of unknown origin (FUO) [140].

Muscle weakness, which is common in PHPT, is effectively though not always definitively reversed after PTX [145,146,147,148,149], but not fatigue and sleepiness. The different muscle-associated effects of PTX may be attributable to the different degrees of hypercalcemia and 25-(OH)D, since low 25-(OH)D levels may reduce musculoskeletal performance [142]. Muscle biopsies from PHPT patients showed an increased expression of PTHR/PTHrPR, and their constant stimulation led to an altered gene expression profile consistent with muscle fatigue [150]. Recently, it has been demonstrated in mice models that PTH receptors initiate the pathway responsible for the development of muscle loss. PKA stimulation results in adipose tissue browning and the expression of atrophy-related genes [151].

Neuropsychological complications are classical manifestations of PHPT. They may be related to the critical action of Ca^2+^, as a secondary messenger in neurotransmitter release [152], changes in monoamine turnover in the central nervous system (CNS), and, possibly, to the direct action of PTH in the CNS through the induction of intracellular Ca^2+^ overload and apoptosis or the reduction in regional cerebral blood flow [142]. Patients with mild PHPT usually complain of depression, anxiety, fatigue, reduced memory, and concentration, but whether and how symptoms improve following PTX has not been determined [103]. Gastrointestinal symptoms in patients with PHPT may arouse suspicion of pancreatitis and peptic ulcer disease [153,154].

Secondary hyperparathyroidism (SHPT) is related to chronic hypocalcemia, which stimulates PTH secretion. The most frequent causes of SHPT are vitamin D deficiency and calcium deficiency. Hypovitaminosis D owing to poor sun exposure, nutritional deficiency, and gut malabsorption is an endemic condition. Impaired active vitamin D function, associated with chronic kidney diseases, is also frequent. Chronic kidney diseases are associated with parathyroid gland hyperplasia, normocalcemia, and elevated PTH levels; when hyperplasic parathyroid cells acquire definitive resistance to the inhibitory effect of [Ca^2+^]_e_ on PTH secretion, hypercalcemia develops and the disease is known as tertiary hyperparathyroidism. In CKD-related SHPT, phosphatemia increases with disease severity. Serum calcium and phosphate levels remain normal until the late stages of CKD at the expense of FGF23 [155].

Symptoms of SHPT are caused by altered bone metabolism, in which elevated PTH and FGF23 levels, low/normal calcemia, and low 1α,25-(OH)_2_D variably contribute to the bone alterations defining the chronic kidney disease-mineral bone disease (CKD-MBD). In women with SHPT due to vitamin D deficiency, independent of 25(OH)D levels, mild to moderately elevated PTH levels are associated with adverse effects on muscle strength and postural stability [156]. Skeletal muscle dysfunction is a clinical manifestation of CKD. The combination of mild to moderate CKD and vitamin D deficiency was significantly associated with hyperparathyroidism and sarcopenia in a geriatric population [157].

## 5. Regulation of PTH Secretion and PTH-Dependent Calcium-Phosphate Metabolism During Exercise

Besides the overall positive health effects derived from regular exercise, bone and muscle benefit from a physically active lifestyle. Different types of exercise training (endurance vs. resistance, aerobic vs. anaerobic, continuous vs. intermittent, and low-moderate-high intensity and volume) elicit different kinds of response by the body [158,159]. Adaptation to exercise results from the multilevel integration of first (direct response), second, and third level (response to soluble factors released by other tissues) mechanical, endocrine, metabolic, and inflammatory responses that involve all organs. There is a difference between the adaptation response to acute exercise and to training, with the latter requiring metabolic shifts and changes in cell function [160]. Research into the effects of exercise on PTH expression and secretion is limited, whereas it is known that calcium homeostasis, and the PTH response as a consequence, during exercise is perturbed and that its maintenance is associated with sustained energy costs; moreover, it has been demonstrated that exercise can modify PTH levels.

### 5.1. Effects of Acute Exercise

Changes in circulating PTH levels during and after acute exercise are duration- and/or intensity-dependent; it has been hypothesized that the PTH response is activated when a threshold in intensity and/or duration is exceeded. In general, marked increases in PTH are observed only for high-intensity and long-lasting (15% above the ventilatory threshold (+15% VT) for more than 50 min) or for low-intensity and very long-lasting (50% VO_2_ max for ≥ 5 h) exercise, whereas short-lasting (30 s) exercise bouts at maximal intensity or long-lasting low-intensity (−15% VT for 50 min) do not impact on PTH secretion [161].

The hypothesis for the existence of such a threshold was advanced by Maïmoun et al. who noted that circulating PTH concentrations were increased after a bout of strenuous exercise in elderly men and women and that this increase might be associated with an anabolic effect on bone metabolism [162]. They reported that, in young adult male cyclists, a 50-min cycling test induced an increase in PTH concentration, as measured at the end of the test and during recovery, only when performed at +15% VT (but not at −15% VT), with no change in serum calcium concentrations [163]. The duration independence was demonstrated by Bouassida et al., two high-intensity submaximal exercise protocols, one continuous (two sequential running trials of 21 min each at 75% and 85% of VO_2_ max, respectively) and the other intermittent (same runs with a 40-min recovery interval in between) elicited an increase in PTH concentration, along with a decrease in ionized calcium concentration in 12 healthy males [164].

Pioneering studies have demonstrated exercise-associated changes in calcium and PTH concentrations after a single bout of various kinds of exercise. Endurance running (45 min, 45% VO_2_ max) reduced plasma ionized calcium measured at 1 and 72 h post-exercise and increased serum PTH measured at 24 and 72 h post-exercise in young women; since no change in PTH occurred in the first 24 h post-activity, the authors suggested that [Ca^2+^]_e_-PTH feedback was preserved in this phase [165]. Similarly, long-lasting moderate-intensity endurance exercise (5 h pedaling at 50% VO_2_ max) lowered ionized calcium and increased serum PTH, though only in the last part of the test [166]. Ashizawa and coworkers demonstrated that strenuous endurance exercise increased the excretion of calcium from the kidney due to decreased tubular reabsorption independent of osteoclastic activity and that this was associated with metabolic acidosis [167]. Additionally, physically demanding, temporarily prolonged work, such as military field maneuvers, was found to be associated with increased PTH concentration, despite the stability of serum calcium and the greater rise in PTH with increasing intensity of physical activity [168]. The alternation of short-duration low-intensity exercise (10 min, 30% VO_2_ max) and equivalently long bouts at increasing-intensity until exhaustion increased PTH concentration in an intensity-dependent manner, which remained elevated for 24 h despite the recovery of serum calcium [169].

The dependency on intensity has been confirmed by other studies. For instance, in endurance-trained runners who performed a prolonged run at either constant (50 min at 4.2 m/s) or increasing intensity (5 steps of 8 min with a +0.25 m/s increase per step), a marked increase in serum PTH despite a rise in calcium concentration was noted in the runners who ran at a constant rate, whereas the changes in PTH did not correlate with the total serum calcium but rather were associated with the rise in the plasma lactate test in the other runners [170]. The 24-h persistence of elevated PTH concentrations after high-intensity exercises has been associated with the bony anabolic effect of PTH [170,171]. Twenty-seven well-trained male endurance athletes underwent an incremental exercise test until exhaustion after a warm-up of 10 min at 2 W/kg. The test was begun at 2.5 W/kg and continued with increments of 0.5 W/kg every 10 min until exhaustion. Hydration was maintained. While Mg^2+^ plasma concentrations did not change during the exercise, stress hormone levels (epinephrine, norepinephrine (NE), glucagon, and cortisol), and PTH were all increased in an intensity-dependent manner. Insulin was decreased, whereas calcitonin was unchanged. During recovery, catecholamines and insulin returned to basal levels [172].

In mice submitted to a single bout of treadmill running, a twofold increase in circulating PTH levels was observed at 30 min after exercise. The mice were pretreated for 21 days with either PTH(7-34) to inhibit PTH signaling, PTH(1-34) to increase PTH signaling, or a vehicle. In the vehicle-treated mice, exercise alone increased trabecular bone volume, without cortical involvement, and adaptation of bone to a more plate-like structure. PTH(7-34) inhibited this adaptation. Treatment with PTH(1-34) increased both trabecular and cortical bone volume [173]. PTH signaling through PTHR1 in the cells of the osteoblastic lineage seemed to mediate exercise-dependent PTH-mediated anabolic effects, as demonstrated in mice conditionally KO of PTHR1 in osteoblasts [174].

Incremental cycle ergometer (beginning at 2.0 W/kg with increments of 0.5 W/kg every 10 min until exhaustion) in well-trained males demonstrated an increase in circulating PTH levels, which became significant at 4.0 W/kg, with a rising trend observed also in the immediate post-exercise period (about 3-fold at 7 min after exercise compared to the pre-exercise level). A similar trend was observed for aldosterone, thyroid-stimulating hormone (TSH), glucagon, glucose, lactate, and non-esterified fatty acids (NEFA), whereas an inverse trend was noted for insulin. PTH levels during the trial correlated with Zn^2+^ but the authors did not measure Ca^2+^ [175]. Similarly, Townsend et al. found that circulating PTH concentrations decreased with the onset of exercise, then increased in an intensity-dependent manner, with the highest levels recorded at the end of exercise performed at 75% VO_2_ max and at 5–7.5 min after exercise. Phosphate levels paralleled the PTH changes, as did ionized calcium, though with an inverse pattern; PTH concentrations negatively correlated with Ca^2+^ levels across all intensities [176].

Other studies failed to demonstrate any effect of exercise on PTH levels, despite increased ionized or total calcium; factors such as type of exercise, intensity, duration, duration of recovery, and modality could be the reason for this failure; for instance, after 50 min cycling at −15% VT in male cyclists [163], 50 min running at 3.3 m/sec in male firefighters [170], after 30 s of modified Wingate test exercise at maximal intensity in male athletes [177], after 2 min-intense one-leg isokinetic workout in healthy male subjects [171]. Plyometric exercise (144 jumps) in peripubertal and young men did not cause any change in PTH, although it was slightly increased in the immediate post-exercise phase before returning to baseline during the 24-h recovery. Sclerostin, the osteocyte-derived negative regulator of bone anabolism, was increased more in young men than in boys, despite the known inverse relationship between PTH and sclerostin [178]. The unresponsiveness of PTH following a single bout of plyometric exercise, despite the change in sclerostin and C-terminal cross-linked type I collagen (CTx-I), a marker of bone resorption, was reported also by Guerriere et al. [179].

The exercise-associated rise in PTH is dependent upon mediators released consequent to the activity and was first identified in catecholamines, in cows, [180] but also acidosis [161]. The dependence on serum ionized calcium variations is controversial, though a certain degree of uncoupling has been described [161]. The effect of adrenergic system activation during exercise is known [181]. More recently, the interdependence between PTH effectiveness on bone and NE release has been demonstrated. Both PTHR1 and the β(2)-adrenergic receptor (β(2)AR) activate a common cAMP/PKA signal transduction pathway through a heterotrimeric Gs. However, while activation of β(2)AR via the sympathetic nervous system (SNS) results in decreased bone formation and increased bone resorption, daily injection of PTH (1-34) increases trabecular and cortical bone mass. Here, we note that PTH has no osteoanabolic activity. In mice lacking the β(2)AR, PTH had no osteoanabolic effect and both bone formation and bone resorption and, hence, the overall turnover, were suppressed [182]. Additionally, lactate and low extracellular pH have been reported to induce PTH secretion, as observed in animal and human studies [183,184,185], though no hypocalcemia occurs [186].

Based on the assumption that during the recovery phase after endurance exercise (hours) circulating PTH concentrations are lower than the pre-exercise levels and that the pre-exercise levels might be influenced by such factors as circadian rhythm, Scott et al. compared post-exercise PTH concentrations with a non-exercising control group. Using a crossover design, the authors compared a 60-min (at 10:30 a.m.) bout of treadmill running at 65% VO_2_ max with semirecumbent rest. Blood samples were taken immediately before (baseline 10:15 a.m.) and after (11:30 a.m.) exercise and during recovery (12:30 a.m., 1:30 p.m., and 2:15 p.m.), in 10 physically active men. Circulating PTH concentrations were steeply increased and higher than in the controls immediately after the exercise. In the exercising group, PTH levels returned to baseline during the recovery phase, although lower compared to the controls. In contrast, albumin-adjusted calcium concentrations were unaffected by exercise, whereas in the exercising group, compared to the controls, phosphate levels were higher immediately after exercise and lower in the second part of recovery [187]. Table 2 presents the main results from studies investigating the effects of acute exercise on PTH and related parameters in humans.

Research investigating the effects of calcium and/or vitamin D supplementation on PTH secretion and mineral metabolism has almost always involved bone metabolism. Shea et al. investigated the effect of vigorous walking (treadmill walking at 75–80% VO_2_ max) on PTH and bone metabolism in elderly women and whether the timing of calcium supplementation influenced this response. Ten women (aged >60 yr) consumed 125 mL of either a calcium-fortified (1 g/L) or a control beverage every 15 min during exercise starting 60 min before and during 60 min of exercise. Serum ionized Ca^2+^ decreased in the control group and remained unchanged in the supplemented group. PTH increased after exercise in both groups, although in an attenuated fashion in the supplemented group. CTx-I increased only in the control group. In a second setting, 23 women of similar age consumed 200 mL of a calcium-fortified (1 g/L) or a control beverage every 15 min starting 15 min before and during 60 min of exercise. Serum ionized Ca^2+^ decreased in both groups, although to a lesser extent in the supplemented group. PTH and CTx increased in both groups without any difference between them. The authors concluded that Ca^2+^ supplementation could be useful in reducing the bone resorbing effects of acute endurance, high-intensity exercise and that the timing may be pivotal in driving these effects [188].

Since cyclists are noted to have poor bone status [189], Haakonssen and coworkers investigated whether a calcium-rich pre-exercise meal would limit the exercise-induced perturbations in bone calcium homeostasis. In their randomized crossover study, 32 well-trained female cyclists completed two 90-min cycling trials separated by 1 day in between. Two hours before each trial either a calcium-rich (1352 ± 53 mg calcium) dairy based meal or a control meal (46 ± 7 mg calcium) was administered. Blood was sampled pre-trial, pre-exercise, and then immediately, 40 min, 100 min, and 190 min post-exercise. The pre-to-post-exercise increase in PTH and CTx-I was attenuated in the calcium supplemented group (*p* < 0.001) and remained lower during recovery. This strategy may be helpful in limiting bone loss in cyclists [190]. Fifty-one men (age range 18–45 yr) were randomized to receive either 1000 mg calcium supplementation or placebo 30 min before a simulated 35-km cycling time trial. Immediately after the exercise, serum ionized calcium levels were decreased in both groups, although the decrease was greater in the placebo group. The exercise-dependent PTH and CTx-I increase was not attenuated by supplementation. A 30-min interval between supplementation and exercise may not be sufficient to optimize intestinal calcium availability during exercise [191]. The infusion of a calcium gluconate solution in trained cyclists during a 60-min cycling bout prevented the decrease in ionized serum Ca^2+^, whereas the infusion of saline did not increase in exercise-induced PTH and CTx-I were observed in the saline-treated group and attenuated in the calcium supplemented group, while PINP, a marker of bone formation, was increased in both groups [192]. In a recently published double-blind, randomized controlled trial, 40 active males were assigned to receive a vitamin D3 supplement (2000 IU/day) or placebo for 12 weeks. At the end of the supplementation protocol, improved vitamin D status, improved power, VO_2_ max, fatigue perception, and PTH concentrations were noted in the supplemented group, although decreased testosterone levels were also observed [193]. Table 3 presents the main results of studies investigating the effects of acute exercise on PTH and related parameters in humans in relation to calcium and/or vitamin D supplementation.

### 5.2. Effects of Chronic Exercise and Training

Chronic training and physical activity status impact on PTH secretion and its response to exercise. A 6-week endurance training program (1 h/day for 4 days/week at 75–80% maximal heart rate) increased the exercise-induced release of PTH in elderly men. These findings hold importance for evaluating bone status in the elderly and for preventing age-associated osteopenia [194]. Brahm et al. reported an inverse relationship between basal serum PTH concentrations and VO_2_ max in men and women (*n* = 10 each) with a wide range of physical ability; the findings suggested that trainability (the gain in VO_2_ max during training) decreases resting PTH secretion [169].

A recent study compared the seasonal effects (fall vs. spring) of discipline-specific training conducted either indoors or outdoors on bone indices and vitamin D status in British university athletes [195]. The circannual variation in 25-(OH)D, the main index of 1α,25-(OH)_2_D status [196], although associated with an ultraviolet (UV) light dose, was modified by lifestyle habits [197]. Cortical area and trabecular density were lower in autumn in the indoor athletes than in the outdoor athletes, while it did not approach significance during spring. Bone mineral density (BMD) and bone mineral content (BMC) at the distal (4%) and proximal (66%) radius were also lower in the indoor group, with no association with the vitamin D status. Serum 25-(OH)D was lower in spring than autumn and it was not dependent upon the training modality (indoor vs. outdoor). PTH was negatively associated with the vitamin D status in the combined groups and the indoor group; the association was more significant in the outdoor group during the spring term. This study clearly indicates that, besides the exercise-dependent shift in PTH secretion, the exercise-dependent and exercise modality-dependent changes in the vitamin D status (as well as vitamin D supplements) can also drive PTH secretion [195].

An inverse correlation between 25-(OH)D and PTH in exercising subjects has been confirmed by other studies. During a 32-week training program involving military recruits, there was a high frequency (7.2%) of stress fractures that was associated with poor vitamin D status (25-(OH)D < 50 nmol/L) and increased circulating concentrations of PTH [198]. Serum 25-(OH)D and CTx-I levels were increased, whereas body mass, body mass index (BMI), fat mass, and circulating PTH concentrations were decreased in post-menopausal women after a 12-week Nordic walking exercise program preformed in spring [199]. PTH signaling affects osteocyte-mediated lacuna remodeling, as observed in the tibiae of 19-week old male mice submitted to treadmill running training for 3 weeks. The perilacunar region displayed a lower mineral-to-matrix ratio and an increased carbonate-to-phosphate ratio in the exercising mice compared to the untrained controls. These changes were inhibited by PTH(7-34), an inhibitor of endogenous PTH signaling, demonstrating that perilacunar remodeling during exercise is dependent on PTH. The exercise-dependent osteocyte response to endogenous PTH was characterized by downregulated sclerostin expression and increased FGF23 expression [200]. In men with low bone mass, serum sclerostin concentrations were decreased as a result of the skeletal load after 12 months of either resistance training or jump-based interventions, whereas the anabolic hormone IGF-1 was increased. PTH remained unchanged, however [201].

As demonstrated by Kambas and colleagues, one’s personal history of physical activity also has an effect on PTH resting levels, beyond its effects on bone, even in children. Indeed, PTH levels were lower in premenarcheal girls engaged in a high level of physical activity than in their low-level physical activity peers, together with higher lumbar and hip BMD and hip BMC and bone formation markers (tissue non-specific (ALP) bone-specific alkaline phosphatase (BSAP) serum activity, and *n*-terminal procollagen type I pro-peptide (PINP)), and lower CTx [202]. Interestingly, a low level of physical activity is a recognized risk factor for PHPT, especially when calcium intake is low [203].

PTH “unresponsiveness” observed in certain acute exercise protocols has also been encountered in specific training regimens in previous work by our group. In highly trained elite cyclists sampled before the start, at mid race, and at the end of a 3-week stage race, after adjustment for plasma volume changes, 25-(OH)D, PTH, and total calcium remained stable, while FGF23 was increased by 50% and positively correlated with the indexes of metabolic effort and the decrease in serum phosphate, though only in the first half of the race [204], which was associated with increased bone resorption [205,206]. Similarly, an 8-week repeated sprint training regimen in young active men had no effect on circulating PTH levels, despite a slight decrease in sclerostin, Dkk1, and osteocalcin [207]. Additionally, 13 weeks of military training, despite the improved bone status (improved site-specific BMD, reduced CTx-I), were unable to induce changes in PTH levels or ionized calcium and PINP [208]. In mice models, 6-week training programs (power, endurance, combined, or untrained) increased circulating PTH levels, together with downregulation of anti-anabolic factors in the cortical portion of the tibia but only in the power-trained mice [174].

Energy availability (EA) restriction is frequently encountered in highly trained endurance female athletes and is associated with the so-called female triad: menstrual cycle irregularities (amenorrhea), nutritional disturbances (anorexia), and bone metabolism deregulation (osteopenia/osteoporosis). In their observational study, Papageorgiou et al. applied 5-day treadmill running protocols (70% VO_2_ max) of either controlled (5 kcal/kg of lean body mass (LBM) per day) or restricted (15 kcal/kgLBM per day) EA in participants (men and eumenorrheic women). The exercise energy expenditure was 15 kcal/kgLBM per day, bone turnover was shifted towards resorption (i.e., increased CTx-I and decreased PINP) in the restricted EA regimen group but only in the women. Insulin and leptin were also decreased, while the restriction had no effect on albumin-corrected Ca^2+^, Mg^2+^, phosphate, and PTH or other hormones (sclerostin, IGF-1, triiodothyronine (T3), and glucagon-like peptide-2 (GLP-2)) [209]. Table 4 presents the main results from studies investigating the effects of exercise training on PTH and related parameters in humans.

## 6. Direct and Indirect Effects of PTH on Skeletal Muscle Mass and Function

### 6.1. Effects of PTH on Skeletal Muscle

Our knowledge of the systemic effects of PTH is based on evidence from pathological conditions. In patients with cancer or CKD, body weight loss is due to a defective metabolism and results in reduced skeletal muscle and adipose tissue mass. This condition is known as cachexia; as a metabolic defect it constitutes a negative risk factor for frailty in cancer patients and their survival; it also limits tolerance to anticancer treatment [151]. Cancer-derived PTHrP or PTH hypersecreted due to secondary hyperparathyroidism in CKD [210,211] is responsible for such modifications. In mice treated with Lewis lung carcinoma (LLC) cells, a standard model of cancer cachexia [210,212], cancer-stimulated heat production was unexpectedly found. It was due to PTHrP, which stimulates thermogenesis in adipose tissue [210] by inducing the expression of uncoupling protein 1 (Ucp1) in the mitochondria-enriched brown (BAT) and beige adipose tissue. These findings, together with the CO_2_ production rate and the respiratory exchange ratio, indicated that in the mice bearing LLC, fat was the preferred energy source. PKA activation by PTH, PTHrP, or NE in white adipose tissue (WAT) and BAT induces the expression of thermogenic genes. An increased expression of PTHrP is also related to upregulation of the ubiquitin-proteasome proteolytic system (UPS), which is involved in muscle protein degradation. Hence, stimulation of PTHR1 can mediate the loss of adipose tissue and lean body mass [212,213]. PTH also affects the catabolism of adipose tissue and skeletal muscle in a mouse model of CKD [211]. Secondary hyperparathyroidism is a complication of CKD and leads to defective bone metabolism and imbalanced calcium and phosphates homeostasis, as well as FGF23 expression, all contributing to poor clinical outcomes (e.g., cardiovascular diseases and metabolic syndrome) and increased risk of death in CKD patients [214]. Subtotally nephrectomized CKD mice displayed a 4-fold increase in circulating PTH in association with reduced body weight, BAT, WAT, and muscle mass, and increased oxygen consumption and energy expenditure. The thermogenic (Ucp1) and atrophy-related (muscle RING-finger protein-1 (MuRF-1) and atrogin-1) genes were upregulated; PTH and PTHrP stimulated these mediators by stimulating PKA activity. The conditional KO of PTHR1 in adipose tissue developed no thermogenic response after PTHrP treatment in either WAT or BAT or skeletal muscle wasting. This effect was due to the suppression of induction of MuRF-1, atrogin-1, and myostatin. A further confirmation came from Ucp1, which has been found upregulated in subcutaneous and deep cervical fat specimens from patients undergoing thyroidectomy for benign illnesses [210,211].

PTH hypersecretion can be found in insulin resistance, together with metabolic acidosis, excess glucocorticoid secretion, chronic low-grade inflammation, and excess angiotensin II [215]. Moreover, PHPT patients are at high risk for developing insulin resistance or T2D; PTX improves insulin sensitivity [216]. This is of note, since in T2D neither WAT browning nor Ucp1 induction takes place, whereas muscle wasting is frequent [217]. In patients with moderate-to-severe CKD (stages 2–5), obesity worsens secondary hyperparathyroidism and, regardless of CKD, obesity is associated with hyperparathyroidism [218], increased PTH levels, insulin resistance, and low-grade inflammation but not with WAT browning [217,218,219,220]. Defective insulin signaling associated with insulin resistance and response to PTH or PTHrP are involved in the activation of a proinflammatory response and the release of cytokines. PTH/PTHrP-induced inflammation can trigger insulin resistance: the inflammatory stimulus stimulates phosphorylation of specific serine residues in insulin-receptor substrate (IRS)-1 before it is degraded by the UPS [215] and the reduction phosphorylated-Akt induces the expression of specific E3 ubiquitin ligases that mark proteins for degradation [213,215]. Since inflammation in cancer and CKD develops also in WAT, BAT, and muscle, inflammation-mediated insulin-resistance may be an alternative mechanism that stimulates adipose and muscle tissues loss. In skeletal muscle, inflammation signals activate signal transducer and activator of transcription 3 (Stat3) and CCAAT-enhancer-binding protein delta (C/EBPδ) downstream. Activation of C/EBPδ induces the expression of myostatin expression and results in enhanced expression of atrophy genes (e.g., atrogin-1, MuRF1, and and myostatin) and muscle protein catabolism [217,221]. The muscle-adipose tissue crosstalk is mediated by irisin, a myokine that induces peroxisome-proliferator activating receptor (PPAR)γ-coactivator alpha (PGC-1α) and stimulates Ucp1, raising energy expenditure [222]. However, the molecular nature of this cross-talk is not well understood and weather PTH directly affects the skeletal muscle function is not known. Indeed, PTHR-1 expression has been demonstrated in skeletal muscle cells in terms of mRNA but not in terms of protein [223]. Hence, it is possible that effects exerted by PTH in muscle cells are secondary to the effects on other tissue and/or on PTH-dependent calcium disposal.

### 6.2. Crosstalk Between Myokines and PTH

Muscle and bone are closely linked by bidirectional signals regulating muscle and bone cell gene expression and proliferation [224,225]. PTH is a master regulator of osteoblast function; PTH signaling on osteoblasts may be modulated by factors secreted by muscle cells, the myokines, which are involved in and modulated by physical activity and exercise. Differentiated primary calvarial osteoblasts cultured in myotube-conditioned media from myogenic C2C12 cells showed reduced mRNA levels of genes associated with osteoblast differentiation, among which the gene encoding the PTHR1 receptor [226]. Figure 3 illustrates the interaction between PTH and myokines.

#### 6.2.1. IL-6 Family Cytokines

The interleukin (IL)-6 family of cytokines is found between bone cells, in bone marrow, and in skeletal muscle in healthy conditions and in disorders in which their levels may be locally or systemically elevated. The IL-6 family of cytokines is defined by their common use of the glycoprotein 130 (gp130) coreceptor, a ubiquitously expressed transmembrane receptor subunit capable of intracellular signaling. Members of the family are IL-6, IL-11, leukemia inhibitory factor (LIF), cardiotrophin-1 (CT-1), oncostatin M (OSM), and ciliary neurotrophic factor (CNTF). Each cytokine that binds to gp130 generates specific intracellular JAK/STAT or ERK signaling cascades by forming specific receptor: ligand complexes [227,228]. Osteocyte-targeted deletion of gp130 was found to ablate the anabolic response to PTH in trabecular bone and on the periosteum [229]. This finding is consistent with the ability of PTH to rapidly promote the expression of IL-6 family cytokines and receptors in osteoblasts, including IL-6 and gp130. The effect was explained, at least in part, by a lower level of PTHR1 expression detected in the bones of mice lacking gp130 in osteocytes. PTH action also relies on suppression of the Wnt signaling inhibitors sclerostin and Dkk1. In mice lacking osteocytic gp130, although PTH treatment was still able to promote RANKL expression by osteoblasts, indicating an intact response to PTH, it was unable to suppress sclerostin or Dkk1 transcription [229], suggesting that the anabolic action of PTH in bone is mediated by gp130-dependent cytokines in osteocytes that suppress sclerostin and Dkk1 [227].

#### 6.2.2. Myostatin

Myostatin (growth-differentiation factor 8, GDF-8) is a member of the transforming-growth factor (TGF)-β superfamily [230] and is expressed mainly in muscle. Myostatin inhibits muscle growth [224,231] and it has a role in age-associated skeletal muscle mass loss during aging (sarcopenia) and in metabolic and inflammatory conditions [232]. It is negatively regulated by acute exercise and training (endurance and resistance in overweight and obese men) [160,233]. Numbering among the antagonists of myostatin are follistatin (FST), follistatin-like 1 (FSTL1), and decorin [160,234].

In older men of the cohort enrolled in the STRAMBO study, serum myostatin levels were not associated with serum PTH levels; nonetheless, serum 25-(OH)D and myostatin levels were positively correlated after adjustment for PTH and season. Data for the effect of vitamin D on myostatin expression are limited: in a mouse myoblast cell line, 1α,25-dihydroxyvitamin D decreased myostatin mRNA expression by 90%, while in aged female rats 1α,25-dihydroxyvitamin D did not affect it. However, these data were obtained using a high dose of the active form of vitamin D and cannot be compared to the relationship between circulating 25-(OH)D and myostatin. The mechanism underlying the association between vitamin D and myostatin and its potential physiological role remain elusive [235].

#### 6.2.3. Irisin

Irisin is a hormone-like molecule that is cleaved and secreted by an unknown protease from fibronectin type III domain-containing protein 5 (FNDC5); it is expressed mainly in the skeletal muscle, heart, adipose tissue, and liver [224,236]. It is produced during physical activity in humans and mice [236]. It acts on adipose tissue by inducing browning of white adipocytes, on bone by increasing bone mineral density, and on muscle. Palermo et al. [237] recently demonstrated the existence of crosstalk between irisin and PTH. Treatment with PTH 1–34 induced downregulation of FNDC5 transcripts and proteins in myotubes, while recombinant irisin exposure reduced *PTHR1* mRNA expression in osteoblasts. Moreover, circulating irisin levels were lower in postmenopausal PHPT patients compared to controls [237,238].

#### 6.2.4. Osteocalcin

Osteocalcin is a 49-amino-acid polypeptide. It is the most abundant noncollagenous protein in the bone matrix, accounting for about 15% of the total. It is mainly produced by osteoblasts and is thought to regulate mineralization; it has many hormone-like features [239]. PTH stimulates osteocalcin synthesis and release from osteoblasts, thus likely modulating the endocrine functions of osteocalcin; osteocalcin stimulates glucose uptake by skeletal muscle fibers and skeletal muscle mass. Osteocalcin signaling in myofibers is responsible for most of the exercise-induced increase in circulating IL-6, which is known to promote adaptation to exercise in part by increasing the production of bioactive osteocalcin [240].

## 7. Conclusions

PTH is a key hormone in the regulation of multiple body organs and systems. Calcium and phosphate metabolism is important for whole-body homeostasis; its dysregulation, because of altered PTH expression and/or secretion, manifests clinically with a wide range of abnormalities in which calcium and phosphate metabolism plays a role.

Studies investigating exercise-associated changes in PTH are far from conclusive and a role for PTH in physically activity-associated health benefits has not yet been established. What emerges from the studies reviewed here is that acute exercise mainly results in an increased PTH secretion especially in the late phase of long-lasting exercise and during recovery, regardless of changes in calcium and phosphorous levels. This rise in circulating PTH concentrations seems to be dependent upon the effort spent with moderate-to-high intensity possibly prolonged exercise being more effective in inducing this elevation. However, chronic exercise may help in limiting PTH secretion, though the effect has been found significant in those studies considering older cohorts involved in endurance activities. This specific aspect may be of particular interest since the conditions of deregulated PTH secretion are more often typical of the elderly. The exercise-induced PTH rise seems to be driven only partially by an exercise-induced increase in calcium levels. This item holds interest for pathology, since a slight reduction in circulating PTH levels could be beneficial by virtue of the multiorgan complications associated with PHPT. A low level of physical activity, together with a limited calcium intake, is associated with an increased risk of PHPT. An optimal physically active status coupled with dietary supplementation (calcium and/or vitamin D) may be beneficial for health. However, such supplementation may need to be opportunely combined with exercise in order to be effective. A novel area of knowledge about the exercise-dependent regulation of PTH secretion is the crosstalk between myokines and skeletal muscle-derived hormones, which are the key mediators of the systemic effects of physical activity.

## Figures and Tables

**Figure 1 ijms-21-05388-f001:**
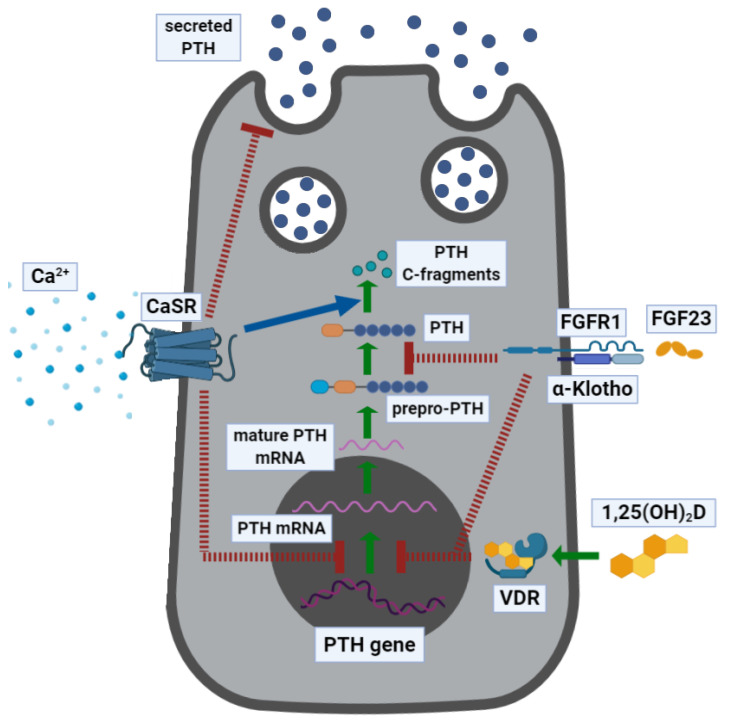
Parathyroid hormone expression and secretion in parathyroid cells. Parathyroid hormone (PTH) is expressed as a prepro-PTH that is then cleaved into mature PTH and stored in granules until secreted. *PTH* mRNA transcription and PTH secretion are inhibited following the activation of the calcium-sensing receptor (CASR) by extracellular calcium [Ca^2+^]_e_, which also stimulates the intracellular inactivation of PTH, into biologically inactive C-terminal fragments, operated throughout the cleavage. Osteocyte-derived fibroblast growth factor (FGF)-23 activates the FGF receptor (FGFR1), heterodimerized with its coreceptor α-Klotho, and inhibits *PTH* mRNA transcription and PTH protein maturation from prepro-PTH. Finally, 1α,25-dihydroxyvitamin D (1α,25-(OH)_2_D) binds the intracellular vitamin D receptor (VDR) and inhibits the expression of *PTH* mRNA. The green arrows indicate PTH expression, the blue arrow indicates a stimulatory pathway, and the red dashed lines indicate inhibitory pathways.

**Figure 2 ijms-21-05388-f002:**
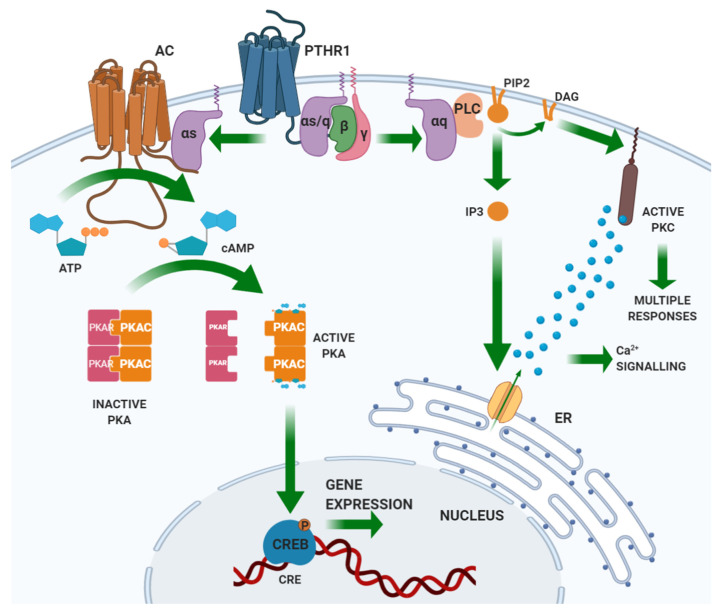
Signaling pathways induced by the activation of the parathyroid hormone receptor. Parathyroid hormone receptor (PTHR)1 is a class II G protein-coupled receptor. Binding of PTH to PTHR1 activates a Gαs protein that activates the adenylate cyclase (AC). AC catalyzes the formation of cyclic adenosine monophosphate (cAMP) from adenosine triphosphate (ATP). cAMP binds and activates protein kinase A (PKA), which, in turn, phosphorylates the cAMP-responsive element binding protein (CREB) into the nucleus, enabling its binding to the cAMP-responsive element (CRE) on DNA and, thus, activates transcription of specific genes. Through the activation of Gαq, PTHR1 activates membrane-associated phospholipase C (PLC), which cleaves the membrane phospholipid phosphatidylinositol-(4,5)-bisphosphate (PIP_2_) into diacylglycerol (DAG) and inositol-(1,4,5)-trisphosphate (IP_3_). IP_3_ diffuses in the cytoplasm, reaches the endoplasmic reticulum, and induces the release of Ca^2+^ by activating receptor-gated Ca^2+^ channels. The Ca^2+^ released from the endoplasmic reticulum activates Ca^2+^-dependent responses and, together with the DAG produced by PLC, activates protein kinase C (PKC), which mediates intracellular responses.

**Figure 3 ijms-21-05388-f003:**
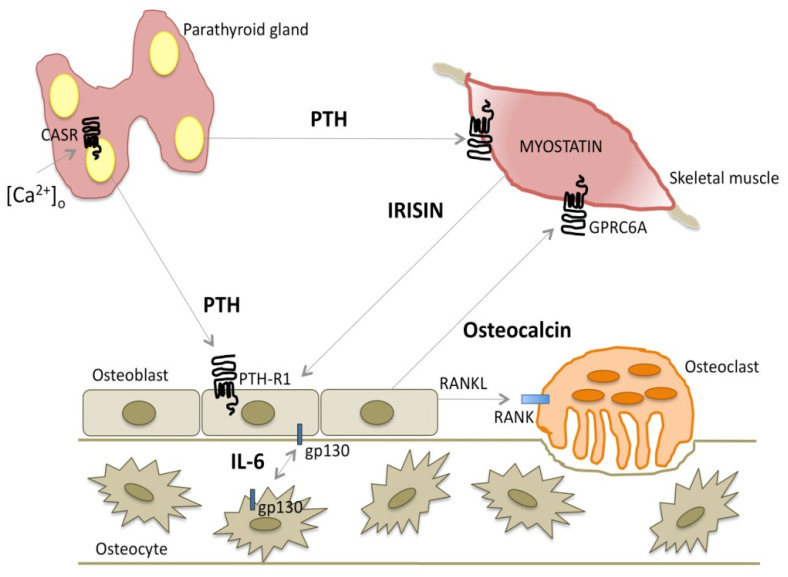
Crosstalk between parathyroid glands, bone, and skeletal muscle. The figure schematically summarizes the main interrelationships between PTH, myokines, and osteokines involved in the crosstalk between parathyroid glands bone and skeletal muscle. Parathyroid-released PTH acts on osteoblasts through the PTHR1 receptor; PTH-stimulated RANKL release activated osteoclasts and osteoclasts-mediated bone matrix reabsorption, participating to bone remodeling, while PTH-stimulated osteocalcin release acts on skeletal muscle cells through the GPRC6A receptor. In addition, PTH itself may act on skeletal muscle cells, which express PTHR1. Besides, in response to physical activity muscle cells secreted irisin, which acts on osteoblasts modulating the expression of PTHR1. Lastly, IL-6, which is also released by muscle cells in response to exercise, mediates a modulatory crosstalk between osteoblasts and osteocytes through the receptor gp130. PTH: parathyroid hormone; PTHR1: parathyroid hormone receptor 1; GPRC6A: G protein-coupled receptor 6A; RANK: receptor activator of nuclear factor κ-B; RANKL: ligand of the receptor activator of nuclear factor κ-B; IL-6: interleukin 6; gp130: glycoprotein 130.

**Table 1 ijms-21-05388-t001:** Characteristics of physical activity and exercise.

**Voluntary Activity**	
**Physical Activity**	Body movement generated by skeletal muscles Variable energy expenditure Positive correlation with physical fitness Life-sustained
**Exercise**	Body movement generated by skeletal muscles Variable energy expenditure Positive correlation with physical fitness Planned, structured, repetitive Aimed at maintaining/improving physical fitness
**Muscle activity**	
**Isometric**	The force generated by the muscle is lower than the external resistance (muscle length unchanged)
**Isotonic**	The force generated by the muscle exceeds the external resistance - Concentric: the muscle shortens - Eccentric: the muscle stretches while the sarcomeres shorten
**Metabolic activity**	
**Resistance**	Short-term power/explosive activity Dependent on anaerobic metabolism
**Endurance**	Mid-to-long-term activity dependent mainly on aerobic metabolism

**Table 2 ijms-21-05388-t002:** Regulation of PTH secretion and PTH-dependent calcium-phosphate metabolism during acute exercise in humans.

Study Cohort	Intervention	Physiological Outputs	Ref.
Endurance Exercise			
Caucasian subjects (*n* = 21; ♂ *n* = 11; ♀ *n* = 10) Age range 60–88 yr Active elderly	Maximal incremental test: - overnight fasting (12 h) - warm up: 5 min at 0% slope - slope increased by 1–2%/min until exhaustion (max test duration 8–12 min) - Blood sampled: - pre-EE - post-EE	Baseline: - iPTH correlated with age (*r* = 0.56), 25-(OH)D (*r* = −20.50) and 1,25-(OH)_2_D (*r* = −20.47). - 25(OH)D and 1α,25-(OH) _2_D correlated with age (*r* = −20.50, *r* = −20.53) - Post-exercise: - iCa, 25-(OH)D: ↓ - iPTH: ↑, correlation with baseline iPTH and age (*r* = −0.46) - 1α,25-(OH)_2_D, CTx-I, OC, BSAP, Hct, (Hb): ↔	[162]
♂ Caucasian (*n* = 7) Age range 20–39 yr Cyclists	2 × 50-min cycling tests: - 15% –VT - 15% +VT - Blood sampled: - pre-EE - during EE (at 30 and 50 min) - post-EE 15 min	- tCa, 25-(OH)D, 1α,25-(OH)_2_D, cortisol: ↔ - iPTH: ↑ in +VT at the of exercise, peak during recovery - P: ↑ in –VT and +VT - Albumin: ↑ in +VT - OC, CTx-I: ↑ in +VT, return to baseline during recovery - BSAP: ↑ in –VT and +VT, return to baseline during recovery	[163]
Caucasian ♂ (*n* = 12) Age range 20–27 yr Healthy	2 EEs (P1, P2) on separate weeks: - P1: 2 × 21-min consecutive exercise at 70% and 85% VO_2_ max; - P2: 2 × 21-min consecutive periods at 70% and 85% VO_2_ max separated by 40 min of rest. Blood sampled: - pre-EE (1 day before and before each protocol) - during EE (min 7 and 21) - end of EE (P1: min 42; P2: min 82) - post-EE (24 h).	- iPTH: ↑ in P1 and P2 and in recovery from P1 - iCa: ↓ during and at end of P1 and P2; return to baseline after recovery - lactate: ↑ during and at end of P1 and P2; return to baseline after recovery	[164]
Caucasian ♀ (*n* = 14) Age 25.2 ± 0.6 yr (mean ± SEM) Healthy, regular menstruations (≥10 cycles in the previous 3 yr)	45-min outdoor jogging for 45, 50% VO_2_ max Blood sampled: - pre-EE (15 min) - post-EE (1, 24, 72 h)	- iCa (PV-adjusted): ↓ post-exercise (1 h, 72 h) - iPTH (PV-adjusted): ↑ post-exercise (24 h, 7 2h). - PICP (PV-adjusted): ↓ post-exercise (1 h), ↑ post-exercise (24 h, 72 h) - ICTP (PV-adjusted): ↑ post-exercise (24 h, 72 h)	[165]
Caucasian ♂ (*n* = 12) Age range 23–42 yr Healthy	5-h bicycle ergometer, 50% VO_2_ max Blood sampled: - pre-EE - during EE (30 min, 1, 2, 3, 4, 5 h)	- iPTH: ↑ within 1 h and remained elevated - iCa, tMg, PO_4_: ↓ - tCa, K: ↑	[166]
Caucasian ♂ (*n* = 17) Age 20.0 ± 0.4 yr (mean ± SEM) 5 month-trained military recruits	7-d, 20/24 h-field maneuver (combat action, forced marches on skis, bivouacking, other winter field actions) Standardized diet (4040 kcal/day) Blood sampled: - post-EE (1 h, 1, 2, 5 d)	- iPTH, tMg, myoglobin: ↓ - tCa: ↔ - P: ↑	[168]
Caucasian (*n* = 20; ♂ *n* = 11; ♀ *n* = 10) Age range ♂ 21–46 yr, ♀ 22–39 yr Wide range of physical capacity	35-min motor-driven treadmill run: - 10-min warm-up, 30% VO_2_ max - 2 × 10 min at successively increased inclination and speed (47% to 76% VO_2_ max) - 4–5-min maximal effort until exhaustion Blood sampled: - pre-EE - during EE (after each submaximal load) - end of EE (after maximal load) - post-EE (30 min, 24 h)	- Htc: ↑ at end of EE, ↓ post-EE (24 h) - Albumin: ↑ during EE (after 76% load) and at end of EE, ↓ post-EE (30 min, 24 h) - OC (PV-adjusted): ↔ - ALP, PICP (PV-adjusted): ↑ during EE (after 76% load) and at end of EE - ICTP (PV-adjusted): ↓ during EE (after 47% load), ↑ at end of EE and post-EE (24 h) - iPTH: ↑ at end of EE and post-EE (30 min, 24 h) - tCa: ↑ at end of EE	[169]
Caucasian ♂ (*n* = 15) Age range 26–36 yr Well-trained (long-distance runners, firefighters), ≥5 yr experience	2 treadmill-running protocols on separate days: - P1: 4-min warm-up, 40 min running with 0.25 m/s increase every 8 min - P2: 4-min warm-up, 50 min running at constant velocity (3.3 m/s–4.2 m/s) Blood sampled: - P1: after 2 and 4 min in warm-up and after each interval during EE - P2: after 2 and 4 min in warm-up and 4 min after EE beginning and every 10 min	P1 - iPTH: ↑ at min 40, and 50 of EE - tCa: ↑ at min 10, 30, 40, and 50^th^ of EE P2 - iPTH: ↑ from 10th to 50th min of EE in long-distance runners - iPTH: ↔ in firefighters - tCa: ↑ from min 4 to min 50 of EE in long-distance runners - tCa: ↑ after 20, 40, and 50 min of EE in firefighters	[170]
Caucasian ♂ (*n* = 27) Age 33.8 ± 6.7 yr (mean ± SD) Well-trained endurance athletes (9.3 ± 3.2 yr experience)	Electromagnetically-braked cycle ergometer: - 10-min warm-up at 2 W/kg - EE at 2.5 W/kg with 0.5 W/kg increments every 10 min until exhaustion Blood sampled: - pre-EE - during EE (at the end of each stage) - post-EE (3, 5, 7 min)	- iPTH: ↑ from exhaustion to end of recovery - Aldosterone: ↑ from +4 W/kg to end of recovery - Calcitonin: ↔ - Adrenaline: ↑from +3 W/Kg to exhaustion - Noradrenaline: ↑ from +2.5 W/Kg to post-EE (3 min) - Dopamine: ↑ from exhaustion to post-EE (3 min) - Insulin: ↓ from +2.0 W/kg to exhaustion, ↑ post-EE to end of recovery - Glucagon: ↑ from +3.0 W/Kg to +4.5 W/kg, ↑↑ from exhaustion to end of recovery - Cortisol: ↑ from +3.5 W/Kg to end of recovery	[172]
Caucasian ♂ (*n* = 18) Age 32.9 ± 5.3 yr (mean ± SD) Well-trained endurance athletes	Electromagnetically-braked cycle ergometer: - 10-min warm-up at 2 W/kg - EE at 2.5 W/kg with 0.5 W/kg increment every 10 min until exhaustion Blood sampled: - pre-EE - during EE (at the end of each stage) post-EE (3, 5, 7 min)	- Zn, TSH: ↑ from +3.5 W/Kg to end of recovery - Se, aldosterone: ↑ from +4.5 W/Kg to end of recovery - Mn, Co, calcitonin: ↔ - iPTH: ↑ from +4.0 W/Kg to end of recovery - Lactate: ↑ from +3.0 W/Kg to end of recovery - Glucose: ↓ from +2.0 W/kg to +3.0 W/kg, ↑ from +4.0 W/kg to end of recovery - NEFAs: ↓ at +2.5 W/kg, ↑ from +3.0 W/kg to +4.0 W/kg, ↑ from exhaustion to end of recovery - Insulin: ↓ from +2.0 W/kg to exhaustion, ↑ during recovery - Glucagon: ↑ from +2.5 W/Kg to end of recovery	[175]
Caucasian ♂ (*n* = 10) Age 23.0 ± 1 yr Healthy, physically active	3 × 30-min treadmill running on separate weeks: - 55%, 65%, 75% VO_2_ max - 2.5 h recovery Blood sampled: - pre-EE - during EE (2.5, 5, 7.5, 10, 15, 20, 25, 30 min) - post-EE (2.5, 5, 7.5, 10, 15, 20, 25, 30, 60, 90, 150 min)	55% VO_2_ max - iPTH: ↓ during EE (5 min), ↑ post-EE (from 2.5 min to 15 min; peak at 5 min), ↓ post-EE (60 min) - PO_4_: ↑ during EE (from 7.5 min to 5 min post-EE; peak at end of EE) - tCa: ↓ post-EE (from 15 min to 30 min) - iCa: ↓ during EE (from 25 min to 90 min post-EE) 65% VO_2_ max - iPTH: ↑ post-EE (from 2.5 min to 25 min; peak at 7.5 min), ↓ post-EE (60 min) - PO_4_: ↑ during EE (from 5 min to 10 min post-EE; peak at the end of EE), ↓ post-EE (from 60 min to 150 min) - tCa: ↑ during EE (from 7.5 min to 5 min post-EE; peak during EE at 20 min), ↓ post-EE (from 25 min to 90 min) - iCa: ↑ during EE (from 2.5 min to 10 min), ↓ post-EE (from 2.5 min to 30 min) 75% VO_2_ max - iPTH: ↓ during EE (5 min), ↑ post-EE (from end of EE to 25 min; peak at 5 min), ↓ post-EE (60 min) - PO_4_: ↑ during EE (from 5 min to 15 min post-EE; peak at the end of EE), ↓ post-EE (from 60 min to 150 min) - tCa: ↑ during EE (from 2.5 min to 5 min post-EE; peak during EE at 20 min) - iCa: ↑ during EE (from 2.5 min to 7.5 min), ↓ post-EE (from 2.5 min to 30 min) - iPTH: no main effect of intensity - PO_4_: no main effect of intensity - tCa (albumin-adjusted): no main effect of intensity - iCa: no main effect of intensity	[176]
Caucasian ♂ (*n* = 7) Age range 19–26 yr Athletes (ice hockey players)	Maximal cycle ergometer EE: - 1-min warm-up - 30-s maximal work - 1-min free pedaling Blood sampled: - pre-EE (1 h) - post-EE (5, 60 min)	- iCa, lactate, [Hb], Htc: ↑ post-EE (5 min) - tCa, iPTH, OC, PICP, ICTP: ↔	[177]
Caucasian ♂ (*n* = 10) Age 26 ± 5 yr (mean ± SD) Healthy, physically active	Controlled diet Cross-over study: - 5-min warm-up at 50% VO_2_ max, 60-min treadmill running, 65% VO_2_ max - semirecumbent rest Blood sampled: - pre-EE (15 min) - end of EE - post-EE (1, 2 h, 2 h 45 min)	Experimental - PTH, PO_4_: ↑ at end of EE (> rest) - Ca (albumin-adjusted): ↔ Control - PTH: ↔ - PO_4_: ↔ - Ca (albumin-adjusted): ↔ Experimental vs. control - PTH, PO_4_: higher at end of EE, lower post-EE (2 h) - Ca (albumin-adjusted): ↔	[187]
**Resistance Exercise**			
Oriental ♂ (*n* = 10) Age 24.3 ± 0.9 yr (mean ± SD) Recreationally adapted	Standardized diet (840 mg/day calcium) 45-min RE program (3 × 7 exercises: bench press, back press, arm curl, double-leg extension, bent-leg incline sit-up, lateral pull down, leg press. - 1st set: 60% 1RM - 2nd, 3rd sets 80% 1RM Blood sampled: - pre-RE (15 min) - end of RE - post-RE (15 min, 45 min, 1 h 45 min, 2 h 45 min) Urine sampled: - pre-RE (30 min-collection) - during RE (1 h-collection) - post-RE (3 × 1h-collection)	Blood - iCa: ↓ at the end of RE, ↑ 45 min post-exercise - tCa (albumin adjusted): ↑ at end of RE - Albumin: ↑ at the end of RE up to 45 min post-RE - PO_4_: ↓ 15 min up to 2 h 45 min post-RE - Lactate: ↑ at the end of RE up to 45 min post-RE - PTH: ↓ 1 h 45 min post-RE Urine - pH: ↓ during RE - NH_4_, renal net acid excretion: ↑ during RE and 1st collection post-RE - TA-HCO_3_^-^: ↑ during RE, ↓ in 2nd and 3rd collection post-RE - P: ↓ in 1st, 2nd, and 3rd collection post-RE - Fraction Ca excretion: ↑ during RE, 1st and 2nd collection post-RE - uCa: ↑ in 1st and 2nd collection post-RE - udPYR: ↓ in 1st collection post-RE	[167]
Caucasian ♂ (*n* = 5) Age 37 ± 2 yr Ordinarily physically trained	- 2-min warm-up - 2-min one-leg maximal isokinetic work (stretching and bending of knee joint at 30°/s) Blood sampled (brachial vein and femoral vein catheterization): - pre-warm-up (2 min) - pre-RE (0 min) - post-RE (2, 4, 7, 17, 30, 60 min)	Leg - tCa: ↑ post-RE (2, 4 min) - iCa: ↑ post-RE (2, 4 min) - Lactate: ↑ post-RE (2, 4, 7, 17 min) - Htc: ↑ post-RE (2, 4, 7, 17 min) - pH: ↓ post-RE (2, 4, 7, 17 min) Arm - tCa: ↑ post-RE (2, 4 min) - iCa: ↑ post-RE (2, 4 min) - Lactate: ↑ post-RE (2, 4, 7, 17, 30 min) - Htc: ↑ post-RE (2, 4, 7, 17 min) - pH: ↓ post-RE (2, 4, 7, 17 min) - iPTH: concentration ↑ post-RE (30 min); content ↓ post-RE (2 min), ↑ post-RE (30 min)	[171]
Caucasian ♂ (*n* = 29; boys, *n* = 12, young men, *n* = 17) Age boys 10.2 ± 0.4 yr, young men 22.7 ± 0.8 yr (mean ± SD) Healthy	Plyometric exercises: 6 jump stations (3 sets of 8 repetitions, total of 144 jumps, 3-min recovery between sets). Adjustment to participant height (drop jumps: 75 cm for men and 40 cm for boys; hurdle jumps: 40 cm for men and 15 cm for boys) Blood sampled: - pre-RE - post-RE (5 min, 1, 24 h)	Boys - iPTH: ↑ post-RE (5 min), ↓ post-RE (60 min) - Sost: ↔ Men - iPTH: ↑ post-RE (5 min), ↓ post-RE (60 min) - Sost: ↑ post-RE (5 min) No differences between groups	[178]
Caucasian ♂ (*n* = 14) Age range 18–32 yr	Calcium-controlled diet (1000 mg/day) prior to and throughout each study period. Randomized cross-over study: - Plyometric exercise: 5–10 min leg-presses warm-up, 10 sets of 10 repetitions at maximal effort jumps at 40% 1RM maximum with 2 min rest between sets. - Non-exercise control period Blood sampled: - pre-RE - post-RE (12, 24, 48, 72 h)	- iPTH, Sost, Dkk1, BSAP: ↔ - OC: ↑ in RE vs. rest - TRAP5b, CTx-I: ↓ in RE vs. rest	[179]

Abbreviations: ♂: male; ♀: female; EE: endurance exercise; iPTH: intact parathyroid hormone; 25-(OH)D: 25-hydroxyvitamin D; 1α,25-(OH)_2_D: 1α,25-dihydroxyvitamin D; iCa: ionized calcium; CTx-I: C-terminal cross-linked type I collagen; OC: osteocalcin; BSAP: bone-specific alkaline phosphatase; Hct: hematocrit; [Hb]: hemoglobin concentration; tCa: total calcium; VT: ventilator threshold; P: phosphorus; SEM: Standard error of the mean; PV: plasma volume; PICP: pro-collagen type I C-terminal peptide; ICTP: C-telopeptide of type I collagen; tMg: total magnesium; PO_4_: phosphate; K: potassium; ALP: alkaline phosphatase; SD: standard deviation; Zn: zinc; TSH: thyroid stimulating hormone; Se: selenium; Mn: manganese; Co: cobalt; NEFAs: non-esterified fatty acids; RE: resistance exercise; % 1RM: percentage of one repetition maximum; NH_4_: ammonium; TA-HCO_3_^-^: urine titratable acidity minus bicarbonate; uCa: urinary calcium; udPYR: urinary deoxypyridinoline; Sost: sclerostin; Dkk1: dickkopf-related protein 1; TRAP5b: tartrate-resistant acid phosphatase 5b.

**Table 3 ijms-21-05388-t003:** Regulation of PTH secretion and PTH-dependent calcium-phosphate metabolism during acute exercise in humans in relation to calcium/vitamin D supplementation.

Study Cohort	Intervention	Physiological Output	Ref.
Caucasian ♀ (*n* = 33, experiment 1, *n* = 10; experiment 2, *n* = 23) Age 61 ± 4 yr (mean ± SD) Healthy, postmenopausal	Overnight fasting (12 h) Exercise protocol: treadmill walking at 75–80% VO_2_ max Experiment 1 - 125 mL calcium-fortified (1 g/L) beverage - 125 mL control beverage starting 60 min before EE and during 60 min-EE, every 15 min. Blood sampled: - pre-supplementation (60 min) - pre-EE (0 min) - end of EE - post-EE (30 min) In experiment 2 - 200 mL of a calcium-fortified (1 g/L) beverage - control beverage starting 15 min before EE and during 60 min-EE, every 15 min Blood sampled: - pre-EE (0 min) - end of EE - post-EE (30 min))	Experiment 1 Supplemented - iCa: ↔ in supplemented, ↓ in controls (no difference in final concentrations) - iPTH: ↑ in supplemented, ↑ in controls (final concentrations in supplemented < controls) - CTx-I: ↔ in supplemented, ↑ in controls (no difference in final concentrations) Experiment 2 Supplemented - iCa: ↓ in supplemented, ↓ in controls (final concentrations in controls < supplemented) - iPTH: ↑ in supplemented, ↑ in controls (no difference in final concentrations) - CTx-I: ↑ in supplemented, ↑ in controls (no difference in final concentrations)	[188]
Caucasian ♂ (*n* = 51) Age range 18–45 yr Road cyclists	Randomized, double-blind, placebo-controlled study Fasted state 35 km-simulated time trial - Calcium supplement (chewable, calcium-citrate (1000 mg elemental calcium)) - Placebo Administration 30 min before EE Blood sampled: - pre-EE - end of EE - post-EE (30 min)	- iCa: ↓ in supplemented, ↓ in placebo (decrease in placebo > than in supplemented) - iPTH: ↑ in supplemented, ↑ in placebo (no difference) - CTx-I: ↑ in supplemented, ↑ in placebo (no difference)	[191]
Caucasian ♂ (*n* = 11) Age range 18–45 yr Cycling-trained	Crossover design Standardized diet, overnight fasting 60-min cycling bouts (3-5-min warm-up before EE, 3–5-min cool-down post-EE) with infusion of - Calcium-gluconate (0.169 mg/mL of elemental Ca) - Saline Infusion started 15 min pre-EE at a rate that delivered 0.5 mg elemental calcium/kg body weight in order to raise serum iCa by 0.1 to 0.2 mg/dL; during exercise the infusion rate was set to deliver 2 mg elemental calcium/min Blood sampled: - pre-EE (start of infusion, immediately before EE) - during EE (every 5 min) - post-EE (15, 30, 45, 60 min, 2, 3, 4 h)	- iCa, tCa (raw and albumin-adjusted): supplemented > controls (all time points post-infusion) - iPTH, CTx-I (raw and albumin-adjusted): supplemented < controls (all time points post-infusion)	[192]

Abbreviations: ♂: male; ♀: female; SD: standard deviation; EE: endurance exercise; iCa: ionized calcium; iPTH: intact PTH; CTx-I: C-terminal cross-linked type I collagen; tCa: total calcium.

**Table 4 ijms-21-05388-t004:** Regulation of PTH secretion and PTH-dependent calcium-phosphate metabolism in relation to exercise training or training status in humans.

Study Cohort	Intervention	Physiological Output	Ref.
Endurance Training			
Caucasian ♂ (*n* = 24) Age range 55–73 yr Healthy, recreationally active	ET protocol: 6 weeks (October to November) bicycle ergometer exercise, 1 h/d, 4 d/week, at 70% HRmax in week 1, 80% HRmax until end of training Test: MET with workload increased by 20 W/min until exhaustion - pre-ET - post-ET	- Maximal aerobic power, maximal oxygen uptake: ↑ in post-ET - PTH: ↑ by MET in pre-ET (correlation with lactate) < than ↑ by MET in post-ET (no correlation with lactate) - OC: ↑ by MET in pre-ET; ↔ by MET in post-ET - tCa, P, ALP: ↑ by MET in pre-ET and post-ET - tCa (albumin-adjusted), 1α,25-(OH)_2_D: ↔ by MET in pre-ET and post-ET	[194]
Mixed (*n* = 47): ♂, *n* = 31; ♀, *n* = 16; indoor (basketball, cheerleading, mixed martial arts, rowing, squash, swimming, table tennis), *n* = 22; outdoor (athletics, American football, football, hockey, lacrosse, rugby, triathlon, frisbee) *n* = 25; Caucasian, *n* = 39; Asian-Indian, *n* = 6; Asian-Chinese, *n* = 2. University-level athletes (training ≥4 h/week)	Observation in October/November and February/March No regular use of sunbeds or sun holidays between October and February. No regular consumption of supplements containing vitamin D. Dietary intake recorded in a 5-d food diary Muscle strength tests: (1) Non-dominant isometric knee extensor strength - 5-min warm up on a cycle ergometer (75 W) - 3 × 5-s maximal contractions of non-dominant leg separated by a 1-min interval (2) Non-dominant handgrip (3 consecutive tests) (3) Counter movement jumps (3 tests) separated by a 2-min rest interval Aerobic fitness: (1) Yo-Yo Intermittent Recovery Test Level 1 (2 × 20 m shuttle runs at increasing speeds, interspersed with 10 s of active recovery)	Autumn - Aerobic fitness, handgrip strength, jump height: indoor = outdoor - Peak knee extensor isometric strength: outdoor > indoor - Cortical area, trabecular density: outdoor > indoor - BMC, BMD: outdoor > indoor - 25-(OH)D: outdoor = indoor - PTH: negative correlation with vitamin D in the combined groups and the indoor group Spring - Aerobic fitness, handgrip strength, jump height: indoor = outdoor - Peak knee extensor isometric strength: outdoor > indoor - Cortical area, trabecular density: outdoor = indoor - BMC, BMD: outdoor > indoor - 25-(OH)D: outdoor = indoor - PTH: negative correlation with vitamin D in outdoor group	[195]
Caucasian ♀ (*n* = 10) Age range 53–65 yr Postmenopausal	ET program: 12-week Nordic walking program (March to June), 60 min (5 min warm-up, 50 min walking interval, 5 min cool-down), 3 times per week Blood sampled: - pre-ET - post-ET	- BMI, body weight, fat mass, PTH: ↓ - Glucose, insulin, HOMA-IR, tCa, OC: ↔ - 25-(OH)D, CTx-I: ↑	[199]
Caucasian ♂ (*n* = 9) Young adults Highly-trained professional cyclists	3-week professional cycling stage race (21 days of competition, 2 days of rest) Blood sampled: - pre-race (day −1) - during race (day 12) - end race (day 22)	- iPTH, 25-(OH)D, tCa: ↔ - FGF23: ↑ (end race) - P: ↓ (during race)	[204]
Caucasian (*n* = 22): ♂, *n* = 11; ♀, *n* = 11 Eumenorrheic	Randomized counterbalanced crossover design 2 × 5-day protocols of controlled (45 kcal/kgLBM per day) and restricted (15 kcal/kgLBM per day) energy availability ET protocol: - Daily run on treadmill, 70% VO_2_ max Blood sampled: - pre-ET - post-ET	- CTx-I: restricted > control in women - PINP, insulin, leptin: restricted < control in women - PTH, tCa: no difference	[209]
**Resistance Training**			
Mixed ♂ (*n* = 38) Age range: 25–60 yr Apparently healthy, physically active (≥4 h physical activity/week in the past 24 months) Low BMD of the lumbar spine or hip (>−2.5 SD T-score ≤ −1.0 SD)	Randomized controlled trials Supplementation with calcium (1200 mg CaCO_3_/d) and vitamin D (10 μg vitamin D_3_/d) 12-month RT protocols: - P1: 2 times per week (squats, bent-over-row, modified dead lift, military press, lunges, calf raises) - P2: jumps 3 times per week (jump exercises varied in intensity, direction, single- or double-leg; 40–100 jumps/session) Blood sampled: - pre-RT - during RT (6 months) - post-RT (12 months)	- BMD: ↑ at lumbar spine in P1 and P2 (end of RT), ↑ at hip in P1 (during RT, end of RT) - Sost: ↓ (end of RT), no difference between P1 and P2 - IGF-1: ↑ (end of RT), no difference between P1 and P2 - PTH: ↔ (end of RT), no difference between P1 and P2	[201]
Caucasian ♂ (*n* = 18) Age range 16–32 yr Recreationally physically active	Randomized controlled longitudinal 8-week protocol - repeated sprint training (10-min warm-up, 18 × 15 m repeated sprints with 17 s passive recovery; 6 min total effort) 3 times per week - Normal daily activities Blood sampled: - pre-RT - during RT (week 4) - end RT (week 8)	- Dkk-1: ↓ in RT (during RT); RT< control - Sost: ↓ in RT (end RT); RT< control - OC: ↔; RT< control (during RT) - OPG, OPN, IL-1β, TNFα, leptin, insulin, PTH: ↔	[207]
**Combined Training**			
Mixed ♂ (*n* = 1082) Age range: 16–32 yr Military recruits	32-week military training program (starting between September and July) Blood sampled: - week 1 - week 15 - week 32	- 25-(OH)D: < 50 nmol/L−1 ↑ incidence of stress fractures - PTH weak inverse correlations with 25-(OH)D at week 15 (*r* = −0.209) and week 32 (*r* = −0.214)	[198]
Mixed ♂ (*n* = 43) Age 21 ± 3 yr (mean ± SD) Army infantry recruits	13-week military training	- Total body aBMD, arm aBMD, leg trabecular vBMD, leg cortical vBMD; leg trabecular volume, 25-(OH)D: ↑ - Leg trabecular area, CTx-I: ↓ - Leg aBMD, trunk BMD, trabecular number, trabecular separation, cortical porosity, trabecular stiffness, PINP, PTH, tCa (albumin-adjusted): ↔	[208]

Abbreviations: ♂: male; ♀: female; ET: endurance training; HR: heart rate; PTH: parathyroid hormone; MET: maximal exercise test; OC: osteocalcin; tCa: total calcium; P: phosphorous; ALP; alkaline phosphatase; 1α,25-(OH)_2_D: 1α,25-dihydroxy vitamin D; BMC: bone mineral content; BMD: bone mineral density; 25-(OH)D: 25-hydroxy vitamin D; BMI: body mass index; HOMA-IR: homeostasis model assessment - insulin resistance; CTx-I: C-terminal telopeptide of type I collagen; iPTH: intact parathyroid hormone; FGF23: fibroblast-growth factor 23; PINP: procollagen type I *n*-terminal peptide; SD: standard deviation; RT: resistance training; Sost: sclerostin; IGF-1: insulin-like growth factor 1; Dkk-1: dickkopf-related protein 1; OPG: osteoprotegerin; OPN: osteopontin; IL-1β: interleukin 1β; TNFα: tumor necrosis factor α; aBMD: areal bone mineral density, vBMD: volumetric bone mineral density.

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
