# Peer review of "Physical Activity-Dependent Regulation of Parathyroid Hormone and Calcium-Phosphorous Metabolism"

_ijms, 2020, doi:10.3390/ijms21155388_

Round 1

Reviewer 1 Report

The review is generally well written however some details are lacking. For example it is not clear if the studies are human/mouse? There may be differences between cell types, species, etc.

  1. Calcium and phosporous metabolism in GI cells? Renal cells? Bone cells? The cell types/species etc. should be mentioned in the citations throughout.
  2. Line 58, “The terms……incorrectly used” is synonymous to the first sentence “In popular….misused” (Line 56). Please avoid using the “same meaning” sentences repeatedly.
  3. Line 69, “distinguished in”>>”distinguished as”
  4. Line 113, 114, 117 “increases”>>”increase”
  5. Line 220-229, Reference?
  6. Line 230-231 “….PTH knockout was noted….”. Please modify the sentence to prevent confusion to the potential readers.
  7. Line 272-281 (1st two sentences). Which review? These sentences are repeated time and again since the abstract section. And reference is missing (Line 272-283). Basically, whole para Line 272-281 is the same thing authors repeated that they mentioned in previous sections.
  8. Reference missing Line 450-60)
  9. Line 788 “IL6 is found in bone marrow cells?? >>>”secreted by bone marrow cells” makes more sense?
  10. Review paper generally collects information from published litereatures and the authors put their opinion for future directions, and bridge the gap in current understanding in the field. This is heavily lacking in this paper. Authors put lengthy introduction of il6, myostatin etc in the litereature, however fail to give their input on how these factors could be connected to physical exercise related field and how this factors could be utilized for future research in the field. Among many examples (practically I find impossible to list), this is one example: authors include a sentence “Irisin is lower in PHPT patients compared to controls”. How does this sentence correlate with the theme of this review paper? So what next after finding irisin to be lower in those patients? What is authors opinion on this finding?

Author Response

Dear Editor,

The authors are grateful about the chance of resubmitting a revised version of the manuscript (ijms-860110) as well as the Reviewers for their critical review and valuable suggestions. The authors have taken into account all the suggestions that the Reviewers have made. The authors hope this new version will satisfy the standards of the Journal.

Looking forward to receive your news.

Prof. Giovanni Lombardi

Reviewer #1

  • The review is generally well written, however, some details are lacking.

The authors sincerely thank the reviewer for useful comments and suggestions

  • For example it is not clear if the studies are human/mouse? There may be differences between cell types, species, etc. Calcium and phosphorous metabolism in GI cells? Renal cells? Bone cells? The cell types/species etc. should be mentioned in the citations throughout.

The specification about the specie/species in which the cited study has been performed has been reported, where possible

  • Line 58, “The terms……incorrectly used” is synonymous to the first sentence “In popular….misused” (Line 56). Please avoid using the “same meaning” sentences repeatedly.

The repetition has been deleted. The new sentence is: “The terms and concepts proper to exercise physiology (e.g., physical activity, exercise, and training) are commonly misused.”

  • Line 69, “distinguished in”>>”distinguished as”

Modified accordingly

  • Line 113, 114, 117 “increases”>>”increase”

Modified accordingly

  • Line 220-229, Reference?

Ref #18 has been entered

  • Line 230-231 “….PTH knockout was noted…”. Please modify the sentence to prevent confusion to the potential readers.

Modified in: “The anabolic role of PTH in bone has been demonstrated in mice in which PTH gene knockout (KO) reduced trabecular bone mass…”

  • Line 272-281 (1st two sentences). Which review? These sentences are repeated time and again since the abstract section. And reference is missing (Line 272-283). Basically, whole para Line 272-281 is the same thing authors repeated that they mentioned in previous sections.

This paragraph has been shortened, to avoid repetitions

  • Reference missing Line 450-60

The last sentence of the paragraph has been substantiated by Ref 161

- Line 788 “IL6 is found in bone marrow cells?? >>>”secreted by bone marrow cells” makes more sense?

Modified accordingly

  • Review paper generally collects information from published literatures and the authors put their opinion for future directions, and bridge the gap in current understanding in the field. This is heavily lacking in this paper. Authors put lengthy introduction of IL-6, myostatin etc in the literature, however fail to give their input on how these factors could be connected to physical exercise related field and how this factors could be utilized for future research in the field. Among many examples (practically I find impossible to list), this is one example: authors include a sentence “Irisin is lower in PHPT patients compared to controls”. How does this sentence correlate with the theme of this review paper? So what next after finding irisin to be lower in those patients? What is authors opinion on this finding?

Although the authors acknowledge this section of the manuscript as less sustained by evidences, since it figures out a field of recent investigation, an attempt to satisfy the request of this reviewer has been made. Specifically, where not present, a brief connection between the molecule’s physiologic role and PTH disturbances has been added. Moreover, a short comment has been added at the end of the conclusion section.

Furthermore, about irisin, although discovered in 2012, the measurement of this myokine still suffers of inconsistency. Determining circulating irisin concentration by the means of commercially available kits, indeed, is still not precise. In our manuscript we quote a sentence from a paper by Palermo et al. which stated that “Irisin is lower in PHPT patients compared to controls” based on available data. However, only one paper, by Jedrychowski et al. (doi: 10.1016/j.cmet.2015.08.001), has reported the recommended concentrations of irisin being in the range 3-5ng/ml. Based on these concerns, in our opinion, to extended discussion of irisin concentration in PHPT patients can only be speculative.

Reviewer 2 Report

This comprehensive review is very well written and interesting. I think it may be publisched after minor English language revision.

Author Response

Dear Editor,

The authors are grateful about the chance of resubmitting a revised version of the manuscript (ijms-860110) as well as the Reviewers for their critical review and valuable suggestions. The authors have taken into account all the suggestions that the Reviewers have made. The authors hope this new version will satisfy the standards of the Journal.

Looking forward to receive your news.

Prof. Giovanni Lombardi

Reviewer #2

  • This comprehensive review is very well written and interesting. I think it may be published after minor English language revision.

The authors are sincerely glad about this reviewer’s positive comment.

The text was originally revised by a professional native English-speaker language editor who is specifically skilled in medical writing. Remaining typos have been corrected.

This manuscript is a resubmission of an earlier submission. The following is a list of the peer review reports and author responses from that submission.

Round 1

Reviewer 1 Report

The paper deals with a current and interesting subject matter; it concerns the impact of physical activity/effort-related changes on calcium phosphate homeostasis and parathyroid hormone secretion. The authors state that, until now, studies in the area have been concerned with the relationship between physical exercise and calcium and phosphate imbalance; but endocrine effect has not been studied yet. However, there is no information regarding the type of imbalance and the reason why the authors decided to focus on PTH secretion only. Currently, there is a lot of researches on other hormones (referred to further in the paper). Thus, an analysis of PTH secretion with no concurrent analysis of, for instance, FGF-23 or vitamin D metabolites in disorders affecting calcium and phosphate homeostasis seems incomplete.

The part of publication on physical activity and effort is presented clearly and does not cause major objections.

The fragments on the physiology and pathology of PTH secretion are very thorough – maybe even a bit too extensive. I believe they could be made more concise as this is not the subject matter of the paper. As a nephrologist and endocrinologist, I would like to draw attention to the following:

- PTH secretion can be stimulated not only by low concentration of ionized calcium, but, in certain situations and at the certain concentrations, also by phosphates. Although the authors say later, that Pi modulates PTH secretion, the role of phosphates in PTH stimulation should be more emphasized;

- no mention has been made of changes in PTH secretion or effects with age and in some diseases (e.g. diabetes and hypertension) or modification of PTH secretion in the treatment of these diseases (calcium channel blockers or converting enzyme inhibitors);

- pulsed PTH secretion has not been mentioned either;

- clinical manifestations of hypoparathyroidism may also be present in the presence of PTH receptor resistance (in which case PTH may be normal or even elevated);

- if primary hyperparathyroidism is defined as the third most frequent endocrine disease, what are the first two?

- the authors decided to clinically assess the symptoms of primary hyperparathyroidism. In this case, disease masking should be mentioned: neuropsychiatric symptoms are classified by the authors as the most important, however, diseases of the gastrointestinal tract (pancreatitis, peptic ulcer disease) and polyuria as one of the first symptoms of hypercalcemia should also be mentioned;

- if MEN 1 syndrome is referred to, MEN 2A should also be mentioned;

- secondary and tertiary hyperparathyroidism is not mentioned AT ALL;

- the authors put much more focus on Ca than P.

I have got no objections at all to the part concerning the relationship between physical effort and changes in PTH secretion.

I suggest the summary section should more straightforwardly define the role of physical effort in regulating PTH secretion and emphasize the practical aspect thereof, i.e., whether, according to the state of the art, physical activity-induced changes in PTH secretion are beneficial to health or not.

Author Response

Replies to Reviewers’ comments

Dear Editor,

The authors are grateful about the chance of resubmitting a revised version of the manuscript (ijms-763792) as well as the Reviewers for their critical review and valuable suggestions. The authors have taken into account all the suggestions that the Reviewers have made. The authors hope this new version will satisfy the standards of the Journal.

Looking forward to receive your news.

Prof. Giovanni Lombardi

Reviewer #1

The paper deals with a current and interesting subject matter; it concerns the impact of physical activity/effort-related changes on calcium phosphate homeostasis and parathyroid hormone secretion. The authors state that, until now, studies in the area have been concerned with the relationship between physical exercise and calcium and phosphate imbalance; but endocrine effect has not been studied yet. However, there is no information regarding the type of imbalance and the reason why the authors decided to focus on PTH secretion only. Currently, there is a lot of researches on other hormones (referred to further in the paper). Thus, an analysis of PTH secretion with no concurrent analysis of, for instance, FGF-23 or vitamin D metabolites in disorders affecting calcium and phosphate homeostasis seems incomplete.

The authors sincerely thank the reviewer for useful comments and suggestions. Alterations of the mineral metabolism in the different disorders have been extended, though concisely, to vitamin D and FGF23.

The part of publication on physical activity and effort is presented clearly and does not cause major objections.

The fragments on the physiology and pathology of PTH secretion are very thorough – maybe even a bit too extensive. I believe they could be made more concise as this is not the subject matter of the paper. As a nephrologist and endocrinologist, I would like to draw attention to the following:

- PTH secretion can be stimulated not only by low concentration of ionized calcium, but, in certain situations and at the certain concentrations, also by phosphates. Although the authors say later, that Pi modulates PTH secretion, the role of phosphates in PTH stimulation should be more emphasized;

The authors have met the reviewer’s suggestion and the role of phosphate has been more extensively presented at pages 2 lines 115-127.

- no mention has been made of changes in PTH secretion or effects with age and in some diseases (e.g. diabetes and hypertension) or modification of PTH secretion in the treatment of these diseases (calcium channel blockers or converting enzyme inhibitors);

These aspects have been now addressed in section 4.

- pulsed PTH secretion has not been mentioned either;

A sentence about this aspect has been added at page 2 lines 107-109.

- clinical manifestations of hypoparathyroidism may also be present in the presence of PTH receptor resistance (in which case PTH may be normal or even elevated);

The authors thank the reviewer for this suggestion: pseudohypoparathyroidism has discussed in the corresponding paragraph at page 7 lines 311-317.

- if primary hyperparathyroidism is defined as the third most frequent endocrine disease, what are the first two?

Primary hyperparathyroidism is the first most common endocrine disorders after diabetes and thyroid disorders; this has been better specified in the corresponding paragraph at page 7 lines 356-357.

- the authors decided to clinically assess the symptoms of primary hyperparathyroidism. In this case, disease masking should be mentioned: neuropsychiatric symptoms are classified by the authors as the most important, however, diseases of the gastrointestinal tract (pancreatitis, peptic ulcer disease) and polyuria as one of the first symptoms of hypercalcemia should also be mentioned;

These aspects have been discussed in the corresponding paragraph at page 8 lines 431-432.

- if MEN 1 syndrome is referred to, MEN 2A should also be mentioned;

MEN2A and MEN4 have been cited at page 8 lines 398.

- secondary and tertiary hyperparathyroidism is not mentioned AT ALL;

These aspects have been now addressed at pages 8-9 lines 433-450.

- the authors put much more focus on Ca than P.

The authors have attempted to provide more insights relative to phosphate throughout the manuscript.

I have got no objections at all to the part concerning the relationship between physical effort and changes in PTH secretion.

The authors are glad to have centred this point.

I suggest the summary section should more straightforwardly define the role of physical effort in regulating PTH secretion and emphasize the practical aspect thereof, i.e., whether, according to the state of the art, physical activity-induced changes in PTH secretion are beneficial to health or not.

The authors agree on the need to draw specific conclusions about the effect of exercise on PTH secretion. However, as reported it is difficult to establish a unique direction of the effect. With this aim the authors have produced tables summarizing the main findings of the discussed studies. Moreover, to meet the request of this reviewer, the authors have added a paragraph in the conclusion section.

Reviewer 2 Report

The review article entitled “Physical activity-dependent regulation of parathyroid homone and calcium-phosphorous metabolism“ by Giouseppe Lombardi at al. discusses the role of exercise and training on parathormon (PTH) serum concentration and the PTH-dependent changes in the plasma calcium and phosphate levels. Given the fact that sports in a controlled fashion is good for our health, it is important to correlate this view with physiological and biochemical parameters

  1. The authors first of all provide definitions for the terms physical activity, exercise and training. This is good explained.
  2. The article continues with a summary of the PTH physiology and the calcium-phosphate metabolism. This is good explained, but I have some remarks to this chapter:

# Stimulation of the G protein Galpha12 has also been connected with the activation of phospholipase C (page 4 of the manuscript);

# cAMP does not bind to CREB but to the PKA holoenzyme (page 4 of the manuscript);

# Phosphorylation of CREB does not regulate the DNA binding of CREB, but rather the interaction of CREB binding protein to CREB;

# Calcium ions are no neurotransmitters, but rather function as second messeners (page 9 of the manuscript);

  1. Chapter 5 is the main chapter of the article, according to the title. Here, the authors listed several studies aimed to investigate a relationship between PTH levels and exercise. This article is quite confusing, because the studies were just listed, without any coherence. The studies are difficult to compare, because the exercise timing and duration are different and the number of participants is different. Later, it is discussed whether the study is performed in the automn or the fall. Accordingly, PTH levels are higher, or not, and calcium serum levels change, or not. The authors summarize these results in the “Conclusion“ section: “The studies about the exercise-associated changes in PTH are far to be conclusive and a role for PTH in the establishment of the physically active-associated health benefits has not yet been determined“.
  2. Chapter 6 discusses the effects of PTH on skeletal muscle and function. Despite the title, the authors discuss rather the role of PTH in adiposites. Concerning the role of PTH in muscle, it would be of interest, whether PTH receptors have been identified on muscle cells. The interaction of PTH, PTH receptors with signaling molecules like myostatin, IL-6 etc. is not really proven, which is even acknowledged by the authors (page 16).
  3. The list of abbreviations needs to be in an alphabetic order.

Author Response

Replies to Reviewers’ comments

Dear Editor,

The authors are grateful about the chance of resubmitting a revised version of the manuscript (ijms-763792) as well as the Reviewers for their critical review and valuable suggestions. The authors have taken into account all the suggestions that the Reviewers have made. The authors hope this new version will satisfy the standards of the Journal.

Looking forward to receive your news.

Prof. Giovanni Lombardi

Reviewer #2

The review article entitled “Physical activity-dependent regulation of parathyroid hormone and calcium-phosphorous metabolism“ by Giovanni Lombardi et al. discusses the role of exercise and training on parathormone (PTH) serum concentration and the PTH-dependent changes in the plasma calcium and phosphate levels. Given the fact that sports in a controlled fashion is good for our health, it is important to correlate this view with physiological and biochemical parameters

The authors first of all provide definitions for the terms physical activity, exercise and training. This is good explained.

The article continues with a summary of the PTH physiology and the calcium-phosphate metabolism. This is good explained, but I have some remarks to this chapter:

The authors are glad about the positive comments of this reviewer and have attempted at satisfying all the following requests.

  1. Stimulation of the G protein Galpha12 has also been connected with the activation of phospholipase C (page 4 of the manuscript);

The authors regret to have not found this information in literature. If the authors have found evidences about the involvement of Gα12 in PLC-mediated signalling in some physiological contexts, no information have been found in relation to PTH receptor activation. If this reviewer believes that this information is relevant, the authors ask to the courtesy to indicate the appropriate literature.

  1. cAMP does not bind to CREB but to the PKA holoenzyme (page 4 of the manuscript);

This reviewer is absolutely right; it was a misprint. The authors thank the reviewer for the observation. The error has been corrected.

  1. Phosphorylation of CREB does not regulate the DNA binding of CREB, but rather the interaction of CREB binding protein to CREB;

The sentence has been revised accordingly.

  1. Calcium ions are no neurotransmitters, but rather function as second messenger (page 9 of the manuscript);

Even in this case the authors are grateful to this reviewer for having reported this misprint. The error has been corrected in “critical action of Ca2+, as a second messenger, in neurotransmitter release”.

  1. Chapter 5 is the main chapter of the article, according to the title. Here, the authors listed several studies aimed to investigate a relationship between PTH levels and exercise. This article is quite confusing, because the studies were just listed, without any coherence. The studies are difficult to compare, because the exercise timing and duration are different and the number of participants is different. Later, it is discussed whether the study is performed in the autumn or the fall. Accordingly, PTH levels are higher, or not, and calcium serum levels change, or not. The authors summarize these results in the “Conclusion“ section: “The studies about the exercise-associated changes in PTH are far to be conclusive and a role for PTH in the establishment of the physically active-associated health benefits has not yet been determined“.

As observed by this reviewer, and as stated by the authors in the conclusion section, the findings are not conclusive, mainly due to the impossibility to compare different studies due to the different protocols applied. Consequently, how exercising affects PTH expression and secretion and, vice versa, how PTH secretion affects the exercise performances are far to be defined. In order to ease the readers, what reported in the text has been summarized in tables, for acute exercise, acute exercise in relation with calcium/vitamin D supplementation, and for training.

  1. Chapter 6 discusses the effects of PTH on skeletal muscle and function. Despite the title, the authors discuss rather the role of PTH in adiposites. Concerning the role of PTH in muscle, it would be of interest, whether PTH receptors have been identified on muscle cells. The interaction of PTH, PTH receptors with signaling molecules like myostatin, IL-6 etc. is not really proven, which is even acknowledged by the authors (page 16).

The authors are conscious about this point. In order to be more precise the following has been added, at the end of the section: “However, the molecular nature of this cross-talk is not well understood and weather PTH directly affects the skeletal muscle function is not known. Indeed, PTHR-1 expression has been demonstrated in skeletal muscle cells in terms of mRNA but not in terms of protein (https://www.proteinatlas.org/). Hence, it is possible that effects exerted by PTH in muscle cells are secondary to the effects on other tissue and/or on PTH-dependent calcium disposal.”

  1. The list of abbreviations needs to be in an alphabetic order.

The abbreviation list, previously sorted based on the order of appearance, has been now arranged alphabetically

Reviewer 3 Report

The authors set out to help us understand the effects of exercise on PTH secretion - an eminently worthwhile goal. Although grammar is mostly correct this article is intolerably verbose. The grammar and word-use errors must be corrected. It must be edited and shortened by a medical professional conversant with modern scientific/medical English use. It can readily be significantly shortened with zero loss of information conveyed. As now written the article would be useless to most readers. The abbreviation list would be easier to use if it were arranged alphabetically. I would like to review the ms. only after these gross insufficiencies have been corrected.

Many paragraphs contain more than one theme. These need to be split. 

Abstract:

“ pivotal since their countless functions…”   use of “since” incorrect.

“  PTH is deputed to the control “ use of deputed incorrect.

Example of bad and unacceptable science reporting…” Degradation of PTH is also conducted into the liver, while their clearance is borne by the kidney [13]. Notably, even if the PTH(1-84) is the prevalent bioactive circulating form, all the PTH-derived peptides retaining the N-terminal have been long recognized to explicate biological activities [14] (e.g., PTH(1-34), whose synthetic form, teriparatide, is an effective drug in the treatment of osteoporosis [15]).…”

Regarding above:

 We might be able to guess what “conducted into the liver” means but no one familiar with English would use such construction. “clearance is borne by the kidney” ditto here. This is silly language use. “even if the PTH(1-84) is the prevalent “ is both inaccurate and obscure. PTH amino acid sequence 1 to 84 is the main circulating form of PTH. Also meaning of “PTH(1-84)” will probably be guessed by readers already conversant with the field, a paper like this must not be cryptic, thus “PTH full length amino acid sequence 1 to 84 is the main circulating form...”.

“to explicate biological activities” is again contorted and inaccurate language. I here explicate why this article needs rewriting. The 34 N-terminal amino acids are the biologically active region of the 84-amino acid human PTH and sold as the bone forming enhancing drug teriparitide. 

Fig.1 is very nice and useful to readers. Can the authors make this larger ? As in the text, Fig.1. legend must be re-written. These comments apply to Fig.2 as well. Fig.3 is also quite well done and helpful but again the legend is grossly inadequate. 

I don’t know what a “mixing posture “ is, nor will the usual interested reader.

Many examples of sloppy proofreading need correction, f.ex. “Interleukin (IL)-6 family of cytokines between bone cells, bone marrow, and skeletal muscle in normal physiology and in pathological states where their levels may be locally or systemically elevated. “ That is not a sentence. All such must be corrected.

Author Response

Replies to Reviewers’ comments

Dear Editor,

The authors are grateful about the chance of resubmitting a revised version of the manuscript (ijms-763792) as well as the Reviewers for their critical review and valuable suggestions. The authors have taken into account all the suggestions that the Reviewers have made. The authors hope this new version will satisfy the standards of the Journal.

Looking forward to receive your news.

Prof. Giovanni Lombardi

Reviewer #3

The authors set out to help us understand the effects of exercise on PTH secretion - an eminently worthwhile goal. Although grammar is mostly correct this article is intolerably verbose. The grammar and word-use errors must be corrected. It must be edited and shortened by a medical professional conversant with modern scientific/medical English use. It can readily be significantly shortened with zero loss of information conveyed. As now written the article would be useless to most readers. The abbreviation list would be easier to use if it were arranged alphabetically. I would like to review the ms. only after these gross insufficiencies have been corrected.

The manuscript has been thoroughly revised and checked for grammar and style by a native English speaker and professional translator.

The abbreviation list, previously sorted based on the order of appearance, has been now arranged alphabetically

Many paragraphs contain more than one theme. These need to be split.

Abstract:

“pivotal since their countless functions…” use of “since” incorrect.

“PTH is deputed to the control” use of deputed incorrect.

All these errors have been revised.

Example of bad and unacceptable science reporting…“Degradation of PTH is also conducted into the liver, while their clearance is borne by the kidney [13]. Notably, even if the PTH(1-84) is the prevalent bioactive circulating form, all the PTH-derived peptides retaining the N-terminal have been long recognized to explicate biological activities [14] (e.g., PTH(1-34), whose synthetic form, teriparatide, is an effective drug in the treatment of osteoporosis [15]).…”

Regarding above:

We might be able to guess what “conducted into the liver” means but no one familiar with English would use such construction. “clearance is borne by the kidney” ditto here. This is silly language use. “even if the PTH(1-84) is the prevalent “ is both inaccurate and obscure. PTH amino acid sequence 1 to 84 is the main circulating form of PTH. Also meaning of “PTH(1-84)” will probably be guessed by readers already conversant with the field, a paper like this must not be cryptic, thus “PTH full length amino acid sequence 1 to 84 is the main circulating form...”.

“to explicate biological activities” is again contorted and inaccurate language. I here explicate why this article needs rewriting. The 34 N-terminal amino acids are the biologically active region of the 84-amino acid human PTH and sold as the bone forming enhancing drug teriparitide.

Besides, the revision of the style and, where needed, of the wrong scientific information, the authors have considered these suggestion and have attempted at improving the manuscript in order to make it as much as possible “reader friendly”. The authors hope that this effort has brought to a, improved version of the manuscript.

Fig.1 is very nice and useful to readers. Can the authors make this larger? As in the text, Fig.1. legend must be re-written. These comments apply to Fig.2 as well. Fig.3 is also quite well done and helpful but again the legend is grossly inadequate.

Figure have been enlarged as much as possible. Legends have been revised, however, if the reviewer has any specific comment regarding the information to be included in the figure legends, the author will take it into consideration.

I don’t know what a “mixing posture” is, nor will the usual interested reader.

The term is referred to the coexistence of different signs of postural instability. We agree with the reviewer about the uselessness of this information. The term has been deleted.

Many examples of sloppy proofreading need correction, f.ex. “Interleukin (IL)-6 family of cytokines between bone cells, bone marrow, and skeletal muscle in normal physiology and in pathological states where their levels may be locally or systemically elevated.” That is not a sentence. All such must be corrected.

As reported above, the manuscript has been thoroughly revised and checked for grammar and style. The authors hope that this new version will be more adherent to the requested scientific and stylistic standards.

Round 2

Reviewer 2 Report

The revised review article entitled “Physical activity-dependent regulation of parathyroid homone and calcium-phosphorous metabolism“ by Giovanni Lombardi at al.contains several changes, but the main mistakes/problems remain.

1.

Phosphorylation of CREB does not regulate the DNA binding of CREB, but rather the interaction of CREB binding protein to CBP/p300.

Chapter 5, describing the relationship between PTH levels and exercise, is still confuse. The tables help to understand the studies, but the results of the studies is very different.  The authors summarize these results in the “Conclusion“ section: “The studies about the exercise-associated changes in PTH are far to be conclusive and a role for PTH in the establishment of the physically active-associated health benefits has not yet been determined“.

The problems with Chapter 6 are still there. The authors are intended to discuss the effects of PTH on skeletal muscle and function. Despite the title, the authors discuss rather the role of PTH in adipocytes. In their response to my comments, it is acknowledged that muscles do not express PTH receptors. The interaction of PTH, PTH receptors with signaling molecules like myostatin, IL-6 etc. is pretty much based on speculation.

.

Reviewer 3 Report

This manuscript has been significantly improved. The content is quite valuable as a review of PTH. It reads like a badly written, detailed chapter in a  med. school physiology text.    However it remains oddly verbose with much empty verbiage. It desperately needs a medical editor familiar with medical writing of this century. It was not edited by a physician or medically trainned editor. It needs this.